# Understanding the drivers of marine liquid-water cloud occurrence and properties with global observations using neural networks

Hendrik Andersen[1,2], Jan Cermak[1,2], Julia Fuchs[1,2], Reto Knutti[3], and Ulrike Lohmann[3]

[1]Karlsruhe Institute of Technology (KIT), Institute of Meteorology and Climate Research
[2]Karlsruhe Institute of Technology (KIT), Institute of Photogrammetry and Remote Sensing
[3]ETH Zürich, Institute of Atmospheric and Climate Science, Zurich, Switzerland

*Correspondence to:* Hendrik Andersen (hendrik.andersen@kit.edu)

**Abstract.** The role of aerosols, clouds and their interactions with radiation remain among the largest unknowns in the climate system. Even though the processes involved are complex, aerosol-cloud interactions are often analyzed by means of bivariate relationships. In this study, 15 years (2001–2015) of monthly satellite-retrieved nearly-global aerosol products are combined with reanalysis data of various meteorological parameters to predict satellite-derived marine liquid-water cloud occurrence and properties by means of regionally-specific artificial neural networks. The statistical models used are shown to be capable of predicting clouds, especially in regions of high cloud variability. At this monthly scale, lower tropospheric stability is shown to be the main determinant of cloud fraction and droplet size, especially in stratocumulus regions, while boundary layer height controls the liquid-water amount and thus the optical thickness of clouds. While aerosols show the expected impact on clouds, at this scale they are less relevant than some meteorological factors. Global patterns of the derived sensitivities point to regional characteristics of aerosol and cloud processes.

## 1 Motivation and aim

Clouds and their microphysical properties play a central role in the Earth's radiative budget by increasing the albedo but also by interacting with outgoing thermal radiation, leading to a net cooling effect (Boucher et al., 2013). Low-level marine liquid-water clouds are the cloud type with the biggest net cooling effect; their shortwave signal by far exceeds their longwave signal (Hartmann et al., 1992; Wood, 2012; Russell et al., 2013; Chen et al., 2014). A global increase in the occurrence frequency or cooling properties of marine low-level liquid-water clouds could thus offset some of the greenhouse gas warming (Latham et al., 2008). Thus, a complete understanding of the physical processes that determine marine liquid-water clouds and their properties is critical.

Atmospheric aerosols are essential for the formation of clouds, influencing cloud properties as cloud condensation nuclei. An increase in aerosol particles leads to a higher cloud droplet number concentration, and, assuming a constant cloud water content, to smaller droplet radii. This changes the cloud's radiative properties, as the larger overall droplet surface area increases cloud reflectivity (Twomey, 1977). These changes in droplet number concentration and size are also thought to have ramifications on cloud lifetime (Albrecht, 1989) and cloud vertical extent (Pincus and Baker, 1994). However, these processes are nonlinear (Bréon et al., 2002; Koren et al., 2014; Andersen et al., 2016; Glassmeier and Lohmann, 2016) and dependent on various

environmental conditions that all feature different patterns in time and space (e.g. Loeb and Schuster, 2008; Stevens and Feingold, 2009; Su et al., 2010; Andersen and Cermak, 2015; Andersen et al., 2016).

Even though there have been significant efforts and advances in understanding aerosol-cloud interactions (ACI) over the last decades, the overall scientific understanding is still considered as low (Boucher et al., 2013). This springs from the complexity of ACI and cloud processes themselves, the temporal and spatial scales at which these processes occur, as well as challenges in observing them.

In the satellite observational community, a typical investigative approach to analyze ACI is to directly relate aerosol and cloud observations quantitatively using bivariate statistics, often explicitly considering one or two meteorological variables (e.g. Matsui et al., 2004, 2006; Chen et al., 2014; Andersen and Cermak, 2015). Even though important process inferences have been made on this basis, the limitation of said method set is clearly that the complexity of the processes is not mirrored by the complexity of the statistical method: only selected aspects of the aerosol-cloud system can be analyzed at one time. A multivariate analysis of the relationships between cloud properties and various predictors, including aerosol and meteorological conditions, might be more appropriate for an adequate representation of these atmospheric interactions. In this spirit, this study combines near-global observational and reanalysis data sets as predictors in a multilayer perceptron artificial neural network (ANN) to model near-global marine water cloud occurrence and properties. The ANN is chosen as it is capable of modeling highly nonlinear functions and does not need any assumptions concerning the data distribution (Gardner and Dorling, 1998). The main goal of this study is to identify the main drivers of marine liquid-water cloud occurrence as well as physical and optical properties on a global scale, estimate sensitivities for each predictor, and determine regional patterns therein.

The guiding hypotheses are:

1. Neural networks are capable of skillfully modeling cloud patterns on monthly time scales, and allow for a separation and estimates of the relative importance of aerosol and various meteorological factors.

2. Global aerosol and cloud patterns are not only related at a global scale, but regional patterns exist as well.

3. At the spatial and temporal scales considered here, meteorological conditions are more important for cloud occurrence and properties than aerosols.

## 2   Data and methods

### 2.1   Data sets

The analysis uses 15 years (2001–2015) of nearly global (60°N–60°S) satellite retrievals and reanalysis fields. Monthly averages of level 3 collection 6 products based on measurements by the Moderate Resolution Imaging Spectroradiometer (MODIS) sensor on the Terra platform (Levy et al., 2013) are used for information on cloud fraction (CLF; data set: Cloud_Retrieval_Fraction_1L_Liquid_FMean), cloud-top droplet effective radius (CDR; data set: Cloud_Effective_Radius-_1L_Liquid_Mean_Mean), cloud liquid water path (LWP; data set: Cloud_Water_Path_1L_Liquid_Mean_Mean) and cloud

optical thickness (COT; data set: Cloud_Optical_Thickness_1L_Liquid_Mean_Mean). While cloud microphysics may also be represented by cloud droplet number concentration, its retrieval requires additional assumptions on vertical cloud water distribution, leading to increased uncertainty (Brenguier et al., 2000), especially in non-stratocumulus cloud regions (Bennartz and Rausch, 2017). As this study investigates liquid-water cloud properties globally, CDR is thus used as a more robust proxy, even though it is also dependent on cloud liquid-water content to some extent (Brenguier et al., 2000). To confine the analysis to liquid-water clouds and to reduce measurement uncertainties due to overlying ice clouds, only single-layer liquid-water cloud products are used. One should note that overlying ice-water clouds reduce the single-layer liquid-water cloud fraction, without actually changing the liquid-water cloud fraction below. This scenario would translate to random noise and potentially blur statistical relationships. However, these effects are thought to be minor, as these situations are likely to average out to some extent in the long-term large-scale data sets used in this study. Also, different cloud products where tested for this study that all yielded similar results. Information on aerosol loading as a proxy for cloud condensation nuclei is provided by aerosol index (AI; computed as a product of the aerosol optical depth (AOD; at 0.55 $\mu$m) and the Ångström exponent (0.55 and 0.867 $\mu$m)). While many studies use the AOD as a proxy for cloud condensation nuclei (e.g. Andreae, 2009; Quaas et al., 2009, 2010; Peters et al., 2012; Koren et al., 2012), the AI has often been found to be a superior measure for this quantity (Stier, 2016), as it weights the fine mode stronger than AOD alone (Nakajima et al., 2001). Some constraints of AI are that it can be affected by aerosol swelling due to hydration in humid environments (Loeb and Schuster, 2008), and that the retrieval describes vertically integrated information and not specifically aerosol at cloud base height where cloud condensation nuclei are typically activated (Shinozuka et al., 2015).

Satellite retrievals are combined with reanalysis data sets from the European Centre for Medium-Range Weather Forecasts (ECMWF) for information on meteorological predictors. The ERA-Interim reanalysis provides data for the time since 1979 and is still continued (Dee et al., 2011). Monthly means of mean daily reanalysis data are used for information on various meteorological predictors at selected atmospheric pressure levels. Meteorological determinants may be grouped into information on relative humidity (RH - at pressure levels 950 hPa (Andersen and Cermak, 2015), 850 hPa (Chen et al., 2014) and 700 hPa (Engström and Ekman, 2010)), vertical velocity (W - at pressure levels 950 hPa, 850 hPa (Kaufman et al., 2005; Engström and Ekman, 2010) and 700 hPa (Engström and Ekman, 2010)), boundary layer height (BLH (Painemal et al., 2014)) and lower tropospheric stability (LTS - computed as the difference in potential temperature between 700 hPa and the surface (Klein and Hartmann, 1993; Chen et al., 2014; Andersen and Cermak, 2015; Andersen et al., 2016)). The reanalysis data used features an original spatial resolution of 0.5°x0.5° and is subsequently resampled to fit the MODIS 1°x1° grid.

Typically, clouds form when air cools off, increasing RH. Once supersaturation is reached, water vapor can condense on the CCN. Predictors are selected that are thought to capture this very basic concept well: Vertical velocity and relative humidity are selected as indicators of cloud dynamics and stratification at various pressure levels. CCN are represented by AI, and BLH and LTS describe the large-scale setting. All predictors have been shown to be relevant determinants of liquid-water clouds or their interactions with aerosols in the studies named above. When available, vertically resolved information is preferred to column integrated (e.g. RH at three different pressure levels is preferred to total columnar water vapor), in order to trace processes at various atmospheric levels. While a higher number of reasonable predictors (e.g. sea surface temperature, geopotential height

or horizontal winds as in Norris and Leovy (1994) or Koren et al. (2010)) is likely to marginally increase the skill of the ANN, it would increase model complexity and make interpretation more difficult.

By design the data sets applied in this study average over time and space to specifically study the large-scale changes within the aerosol-cloud-climate system and to allow for future comparisons with global climate models. While on these scales, the causal sequence of cloud processes may not be intact and the processes themselves cannot be observed, their overall ramifications are thought to be represented adequately, in that temporal averaging is intended as a proxy for process relationships.

## 2.2 Artificial neural networks and study design

### Basics of artificial neural networks

Machine learning systems consist of a set of numerical operators designed to compute a designated output on given input data. The basic principles, such as the number of numerical links between parameters, are fixed. Artificial neural networks can be described as a branch of machine learning systems. Multilayer perceptrons are a specific type of neural network that are commonly used in the atmospheric sciences and environmental sciences in general, as they are able to model highly nonlinear functions. This type of ANN consists of several layers of interconnected neurons. In general, the architecture of multilayer perceptron ANNs is variable but a typical ANN may consist of an input layer, at least one hidden layer and an output layer. The information from an input pattern is strictly passed from the input layer via the hidden layer(s) to the output layer that yields the desired output pattern (feed-forward ANN). Multilayer perceptron ANNs are fully connected, i.e. each neuron is connected to every neuron in the neighboring layer(s). All connections between neurons in the ANN are specifically weighted so that the information passed to a neuron is the sum of the weighted outputs from the previous layer (net input). The neuron modifies the information by multiplication with a nonlinear transfer function and passes this information through specific weights to all neurons of the following layer (Gardner and Dorling, 1998).

In general, these types of ANNs learn through training. During the training period a subset of the input and output data sets are fed into the ANN. Using this training data, a learning algorithm adjusts the individual weights of each neuron in the network to minimize the error of the output (e.g. the difference between the modeled and observed outputs). The speed of the learning process is adjusted by a learning rate that determines the step size taken during the iterative learning process. While a high learning rate leads to faster convergence, it may miss a global optimum. An additional momentum term adds a fraction of the previous weight change to the current weight change to assist the optimization algorithm out of local minima (Gardner and Dorling, 1998). After the learning algorithm has reached convergence, the predicted output of the network can be compared to the original output for an estimate of model skill. To ensure that the ANN does not only represent the particular data used in the training (overfitting) and is able to generalize the functional relationships underlying the training data, the model is validated using a second independent subset of the input data. If the ANN is able to generalize the relationships between the data sets, the difference between training and validation errors and the overall error are small. The ANN is tested on a third set of independent data to ensure that the model is not overfit to the validation data.

## Design of the study and application of the neural network

The ability of the ANN to predict cloud occurrence and properties is dependent not only on an informed choice of predictors, ANN also require sufficient data that fully represent all cases that the ANN is required to generalize, as ANNs perform well for interpolation but poorly for extrapolation (Gardner and Dorling, 1998). In order to circumvent sampling issues and to enable

a direct comparability of results in different regions, the near-global data sets are summarized in 40x20 equal area grid cells by aggregating grid cells at the original spatial resolution of $1°x1°$. This leads to an increase from the original 180 data points (15 years, 12 months) for each input/output to between 8,000 and 14,000, depending on the number of $1°x1°$ pixels that fall into a specific region. Region-specific neural networks are needed to capture regionally varying relationships between cloud properties and their determinants. These relationships feature regional patterns as they depend on liquid-water cloud type,

aerosol composition, meteorology and the respective seasonal cycles, all of which exhibit regional patterns (e.g. Stevens and Feingold, 2009; Andersen et al., 2016). These regional characteristics would be blurred or missed completely when using a single global ANN.

The ANN is only applied in grid cells where a minimum of 2000 valid observations exist. In each equal area, an independent ANN is trained over 500 epochs (i.e. number of times the network iterates over the training data) with 60 % of the data,

validated and tested on 20 % of the data each. A simple network topology with one hidden layer consisting of five hidden neurons is applied a) for a more comprehensible model and b) to reduce potential overfitting (Gardner and Dorling, 1998). Multilayer perceptrons with just one hidden layer are frequently used in ecological studies (e.g. Hartmann et al., 2008; Cermak and Knutti, 2009) as they have been shown by several independent studies to be able to approximate any continuous function (Cybenko, 1989; Funahashi, 1989; Hornik et al., 1989; Kecman and Vojislav, 2001; Olden and Jackson, 2002; Di Noia et al.,

2013). A hyperbolic tangent is used as the activation function, the weights are initialized randomly from a uniform distribution between -0.1 and 0.1. Gradient descent (Werbos, 1990; Le et al., 2017) is used as the optimization algorithm, with a learning rate of 0.003 and a momentum of 0.01. In-depth testing was undertaken to adjust the details of the model's settings by comparing model skill for a wide number of model setups as in Hartmann et al. (2016). Once the ANN is trained and able to generalize the relationships between the data sets adequately, sensitivity analyses can be conducted. In general, sensitivities are systematically

tested by varying each input variable while holding all other input variables constant, e.g., at their average (details see below). In this way the individual contributions of each variable can be analyzed (Olden and Jackson, 2002). A schematic view on the general architecture of the ANN and the training, validation and sensitivity steps is given in Fig. 1.

The ANN skill in modeling the desired outputs is evaluated with the correlation ($R^2$) between ANN testing output and the corresponding observation data. Sensitivities are only computed for grid cells, where the ANN $R^2 > 0.5$ and the root mean

square error relative to the mean (rel. RMSE) $<$ its global average in order to only investigate sensitivities of models that are capable of adequately representing the observed cloud patterns. The derived average sensitivities are only valid for the considered regions and should thus not be interpreted as 'global'. In order to derive a representative and meaningful sensitivity, the mean of ANN-predicted outputs are compared for two groups of input data: all retrievals of a specific predictor smaller than its 25th percentile and all retrievals greater than its 75th percentile; in all cases, the other predictors are held constant at their

grid-cell specific mean values. In comparison to a stepwise increase of one specific predictor, a more relevant measure of a

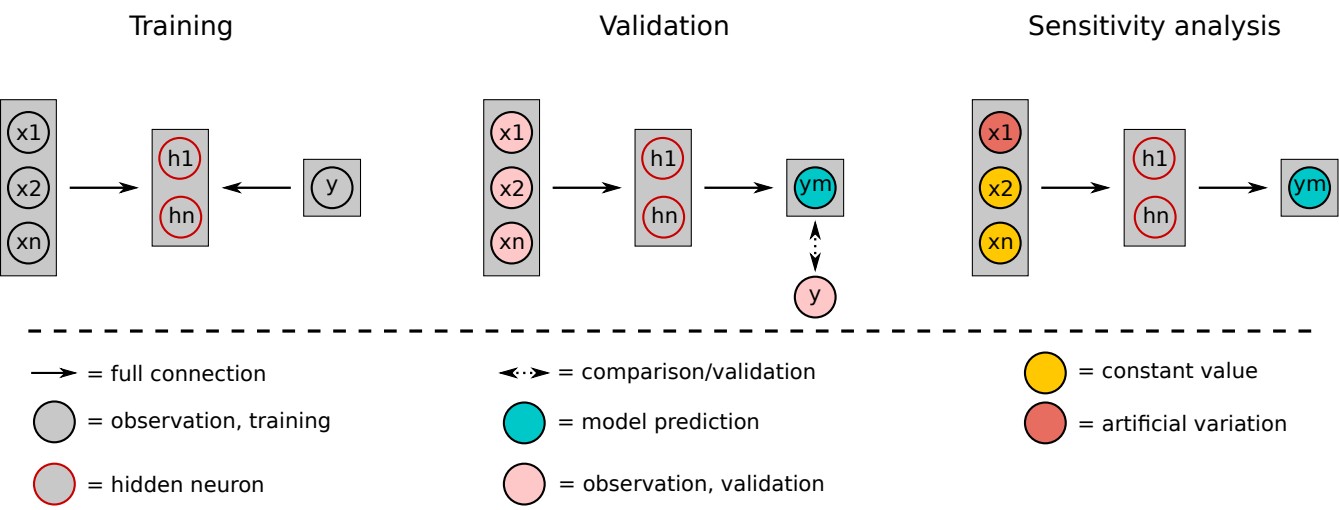

**Figure 1.** A schematic view on the general architecture and design of multilayer perceptron artificial neural networks. In this study, the ANN features a single hidden layer with 5 neurons.

typical sensitivity can be derived, as the predictor distribution is considered. Thus, in the context of this study, the sensitivity is defined as the mean difference between the predicted cloud property in the groups of low and high predictor values. Typically, the aerosol effect on e.g. CDR is described by the $\delta log(CDR)/\delta log(aerosol)$ relationship, where aerosol can be either AOD or AI (e.g. Costantino and Bréon, 2013). While this gives a regionally comparable estimate of the aerosol-cloud sensitivity,

it does not explicitly consider the meteorological framework. As the temporal and spatial scales considered in this study are much larger than the actual processes, the calculated sensitivities represent the system scale, and may not match the magnitude of the sensitivities at the process scale (McComiskey et al., 2009; McComiskey and Feingold, 2012).

## 3 Results and discussion

### 3.1 Skill of the ANN in predicting cloud occurrence and properties

The skill of the ANNs to predict marine liquid-water cloud occurrence, as well as physical and optical properties is shown in Fig. 2 (blue boxes) and contrasted with the skill of a multiple linear regression using the identical set of predictors (red boxes) and a simple Pearson correlation between log(AI) and the cloud properties (black boxes). In the ANN, CLF is predicted with the highest accuracy. While for CDR the skill of the ANN is also $> 0.5$ for many regions, LWP and COT are predicted less accurately. The skill of the multiple linear regression is close to the skill of the ANN, but typically explains a few percent less

of the cloud variability, possibly indicating a small contribution in model skill by nonlinear representations of relationships within the ANN. While the ANN is chosen in this study due to its slightly superior predictive capabilities, figure 2 suggests that other multi-variate methods would have been appropriate as well. It is shown that log(AI) alone typically explains less

than 20 % of the cloud property variability. As a much higher fraction of the cloud variability is explained in the multivariate approaches, the sensitivities derived from the ANN are likely to be more reliable.

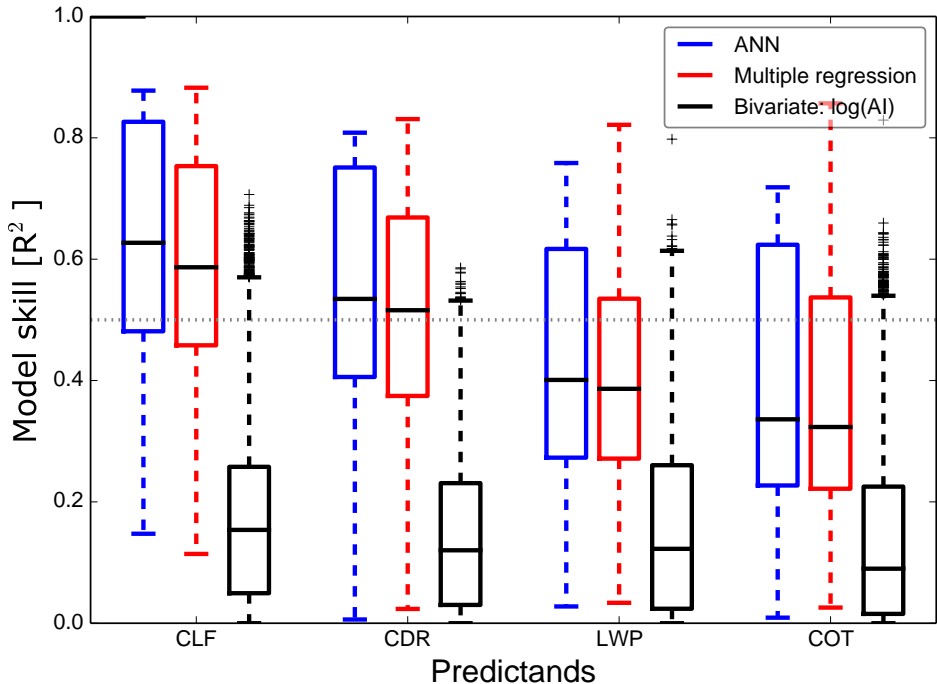

**Figure 2.** Predictand correlation with ANN test output, multiple linear regression (both multivariate) and log(AI) (bivariate). The median is represented by the black horizontal line, framed by the interquartile range (boxes), whiskers expand the boxes by 1.5 interquartile ranges.

For all predictands there is a large spread in model skill, leading to distinct regional patterns as illustrated in Fig. 3. The skill of the ANNs is generally higher in the atmospherically stable regions off the western continental coastlines that are dominated by stratocumulus clouds. Less skilled ANNs can generally be found in the (sub-)tropic Pacific and the Indian Ocean. These regions with comparatively low ANN skill may point to regions where the aerosol-cloud-climate system cannot be sufficiently explained with the choice of predictors used in this study and may thus represent regions of interest for future studies.

The global spatial patterns of ANN skill are likely linked to the spatial patterns of the variability of the specific predictands (Fig. 4). A strong dependence on the variability can be noted for CLF and CDR (Fig. 4a and 4b), i.e. a higher variability enables the ANN to more skillfully represent the inherent relationships. This is sensible, as a higher predictand variability offers the ANN a stronger signal from which it can learn.

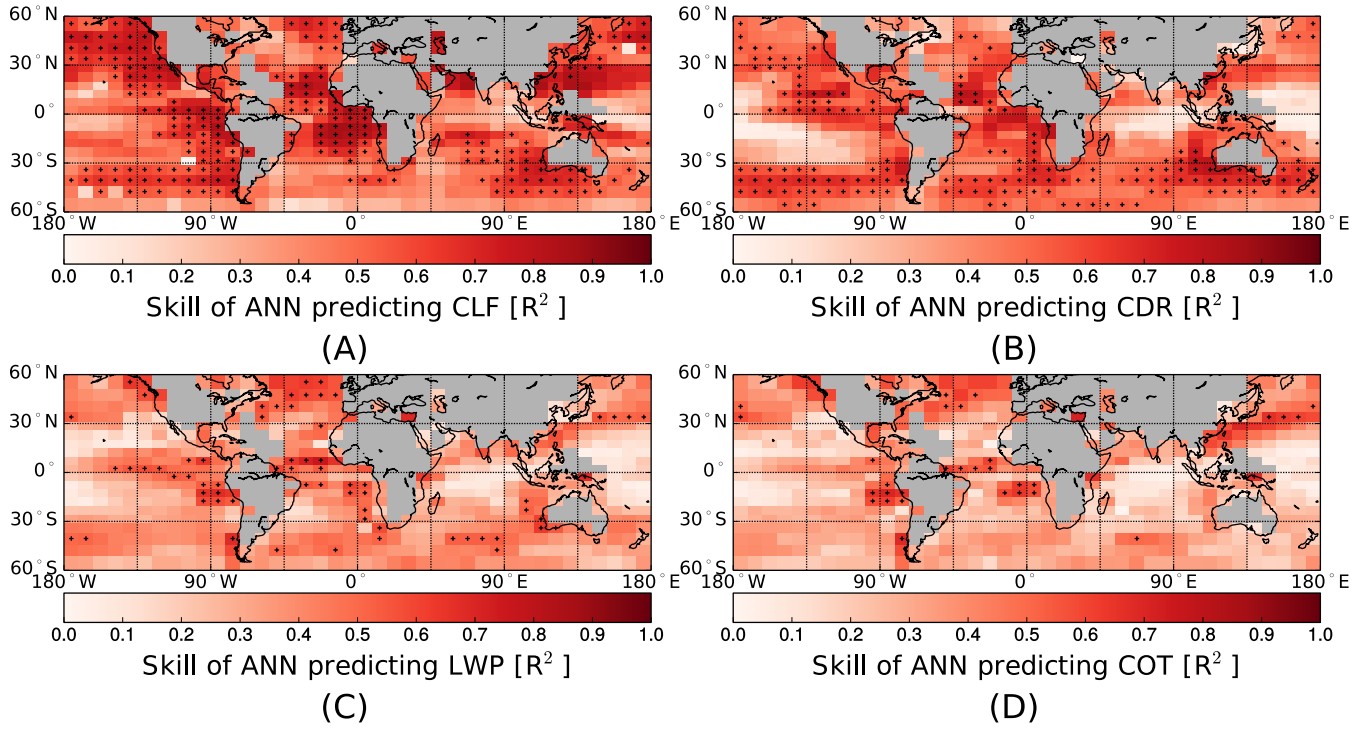

**Figure 3.** Global patterns of ANN skill [$R^2$] predicting a) cloud fraction, b) cloud droplet effective radius, c) cloud liquid water path and d) cloud optical thickness. As only ANNs with $R^2 > 0.5$ and rel. RMSE $<$ its global average are used to compute the sensitivities, these are marked by a '+'.

## 3.2 Determinants of cloud occurrence and properties

Sensitivities are analyzed in all ANNs with a skill of $R^2 > 0.5$ and with a rel. RMSE that is smaller than its global average. Figure 5 shows globally summarized mean and standard deviation of all predictor sensitivities for CLF (Fig. 5a), CDR (Fig. 5b), LWP (Fig. 5c) and COT (Fig. 5d). Positive sensitivities point towards a positive response to an increase in the specific predictor while holding the other predictors constant at their regional average values. CLF shows the greatest sensitivity to LTS, where an increase in LTS leads to a strong increase in CLF, underlining the importance of LTS found in earlier studies (e.g. Klein and Hartmann, 1993; Matsui et al., 2004; Andersen and Cermak, 2015). CLF is also positively related to relative humidity at all assessed pressure levels, with the strongest sensitivity at 950 hPa, where stratocumulus clouds and transitional clouds between stratocumulus and shallow cumulus are located (Gryspeerdt and Stier, 2012; Andersen and Cermak, 2015). While boundary layer height and aerosol are also positively connected to CLF, W sensitivity varies in sign. Sensitivities associated with W can generally be interpreted as the change in the predictand when W changes from updrafts to downdrafts. The most relevant pressure level in terms of W seems to be 700 hPa, with strong positive sensitivities, illustrating that the downdrafts at 700 hPa associated with stable conditions in the lower troposphere correspond to an increase in CLF. In terms of CDR sensitivities, LTS

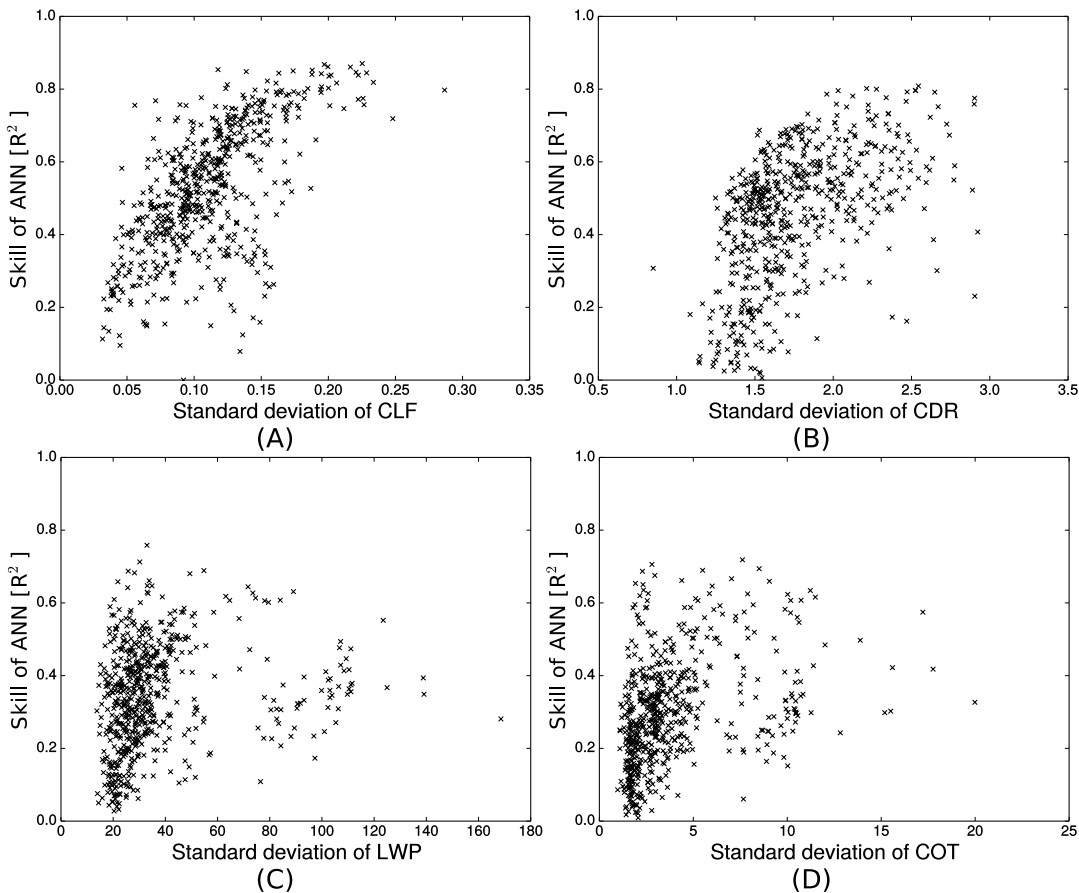

**Figure 4.** The skill of the ANN predicting a) CLF, b) CDR, c) COT and d) LWP as a function of the predictand variability (standard deviation). Each point illustrates the combination of skill and variability for a specific equal area region (pixel regions in figure 3).

also displays the strongest effect, with an increase in LTS connected to a distinct reduction in droplet size. RH at 850 hPa exerts the strongest positive CDR sensitivity, with many of the cloud tops located at this pressure level. AI has a notable sensitivity, showing a distinct negative association to CDR as previously assumed. Generally, updrafts favor larger CDR, with a stronger sensitivity at higher altitudes. Results of LWP and COT sensitivities are similar in terms of the signs and magnitudes of the

5   individual sensitivities. Both are mainly determined by BLH and LTS, both positively associated with the respective cloud property. RH facilitates thicker clouds containing more liquid water, especially free tropospheric relative humidity at 700 hPa seems to have a positive impact on LWP and COT, as higher humidity levels at 700 hPa are likely to weaken drying effects of entraining air masses (Ackerman et al., 2004; Chen et al., 2012, 2014). While increases in aerosol lead to a negative LWP response, this does not lead to a similarly strong COT reduction. W is negatively related to both cloud properties, as situations

10   with updrafts generally produce thicker clouds, the most relevant pressure level is at 850 hPa.

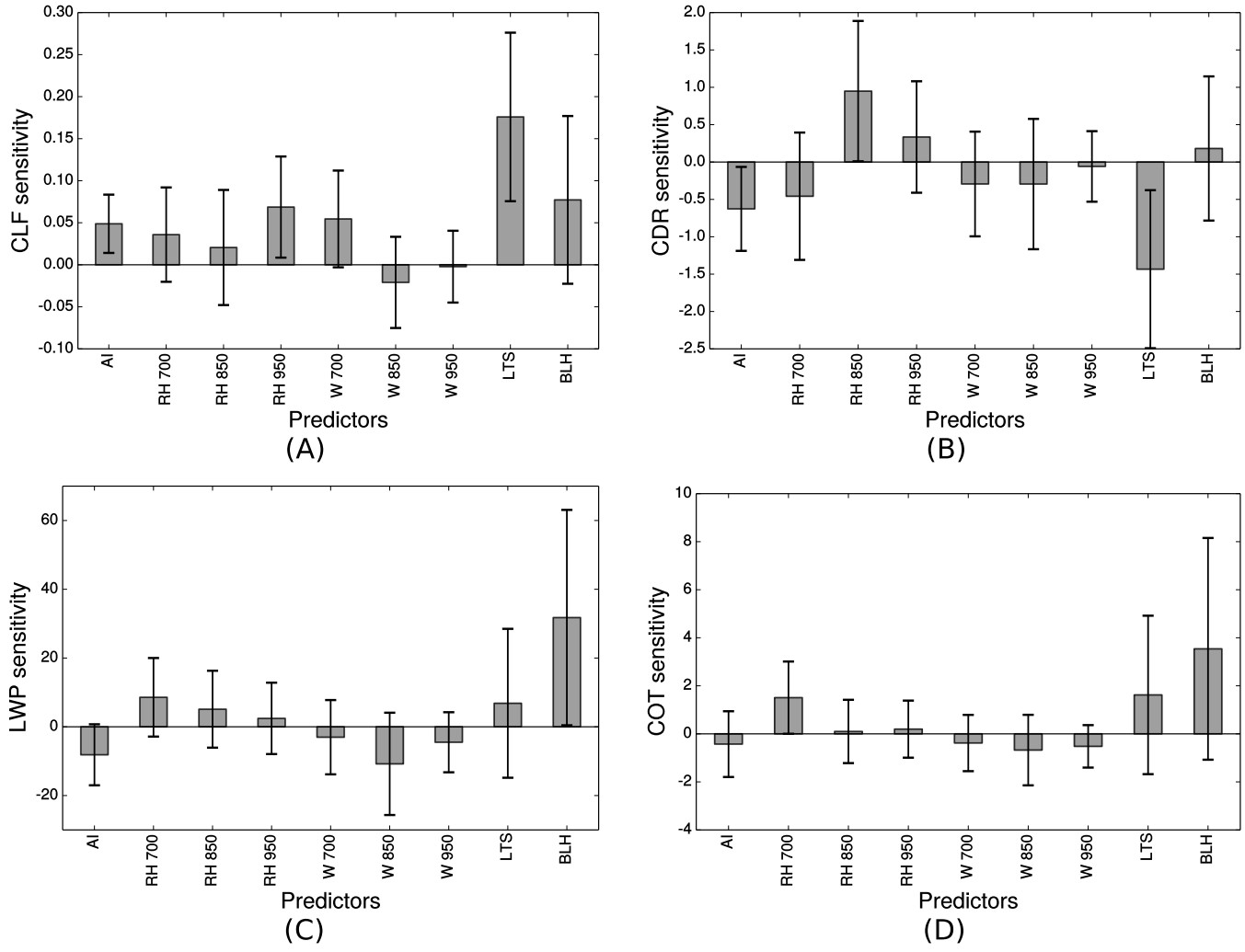

**Figure 5.** Global mean relative sensitivities as defined in section 2.3 of a) CLF, b) CDR, c) LWP and d) COT for all predictors of the ANNs (x axes). Error bars illustrate the regional variability of the sensitivities (global standard deviation).

The application of an individual ANN in every grid cell enables the analysis of regional patterns of the derived sensitivities. Panels of the left-hand column of Fig. 6 show regional patterns of CLF sensitivities, the panels of the right-hand column show regional patterns of CDR sensitivities. The range of the colorbars is identical within each column, so that both the overall magnitude as well as the spatial patterns of the sensitivities can be compared. LTS is the strongest determinant for CLF and is positively related to CLF everywhere on Earth, with especially strong sensitivities in atmospherically stable regions off the western coasts of the continents, where stratocumulus clouds are predominant (Klein and Hartmann, 1993; Russell et al., 2013). In these regions RH shows a strong positive CLF sensitivity at 950 hPa, pointing to the relevance of low-level humidity in these regions of low boundary layer clouds. Liquid water cloud fraction in the intertropical convergence zone is more sensitive to

RH at 850 hPa, reflecting the thicker boundary layer in this region. The most pronounced relation between the aerosol and CLF can be found at latitudes around 30°, especially over the Northwest Pacific.

CDR is markedly reduced by AI in the Northwest Pacific and the Southwest Atlantic and negatively associated with AI to a lesser degree in most other marine regions. The region close to the coastline of the Arabian Sea is an exception. Here, dust particles make up a significant portion of the aerosol species composition (Prospero, 1999; Kaufman et al., 2005), which may lead to larger droplet sizes when dust aerosols act as giant CCN (Levin et al., 2005; Barahona et al., 2010). The Southeast Atlantic features weakly positive sensitivities of CDR to changes in AI. While CDR is typically found to be negatively related to AI in the Southeast Atlantic (e.g. Costantino and Bréon, 2013), Andersen and Cermak (2015) found that AI and CDR can be positively associated in very stable atmospheric conditions. Issues of sampling (few aerosol retrievals in high CLF regions), scale (highly aggregated data) or their combination might affect the observed CDR sensitivity to AI in this region. One should note that sensitivity maps were also produced using AOD as a proxy for cloud condensation nuclei instead of AI. While the overall results were very similar, changes in sensitivity of CDR to the aerosol quantities were observed in the Northeast Altantic that is dominated by Saharan dust aerosols. Here, the difference between AOD and AI is substantial due to the abundance of coarse dust particles. LTS is negatively associated with CDR, especially south of 30° and in the subtropical Atlantic as found by Matsui et al. (2006), as high LTS environments are connected with weaker updrafts and a shallower boundary layer, limiting cloud droplet growth. This excludes the Southeast Atlantic, where stable conditions may trap the humidity in the boundary layer (Johnson et al., 2004; Painemal et al., 2014; Andersen and Cermak, 2015). Similar effects may occur in the Southeast Pacific as well. RH features the strongest positive CDR sensitivity at 850 hPa with distinctly strong sensitivities in the subtropic regions, where cloud tops are frequently located at this pressure level (Gryspeerdt and Stier, 2012). Compared to these factors, W at 700 hPa seems to be a relevant determinant in very selected, mostly tropical regions only.

## 4 Summary and conclusions

The central aim of this study was to identify and analyze the main determinants of marine liquid-water clouds and their sensitivities at the system scale. Artificial neural networks were shown capable of predicting cloud patterns on a global scale well, although ANN skill is dependent on the cloud property and its variability. Regions with a strong monthly variability such as the stratocumulus regions that feature a strong seasonal cycle are most skillfully represented.

Sensitivities were derived for all predictor-predictand combinations, revealing LTS to be the main determinant of monthly liquid-water cloud occurrence and properties. LTS is positively related to CLF on a global scale, with especially strong regional sensitivities in the subsidence regions and the mid-latitudes. In most of these regions, LTS features a strong negative sensitivity towards CDR. One exception to this negative CDR-LTS relationship is the Southeast Atlantic, where high LTS conditions may trap humidity in the boundary layer, causing larger CDR and hence a positve CDR-LTS relationship (Johnson et al., 2004; Painemal et al., 2014; Andersen and Cermak, 2015). The sensitivity of cloud properties to changes in relative humidity is dependent on both region and pressure level. CLF in regions that feature predominantly stratocumulus clouds or other low-level clouds is most sensitive to RH at 950 hPa, whereas tropical regions with thicker boundary layers are more sensitive to

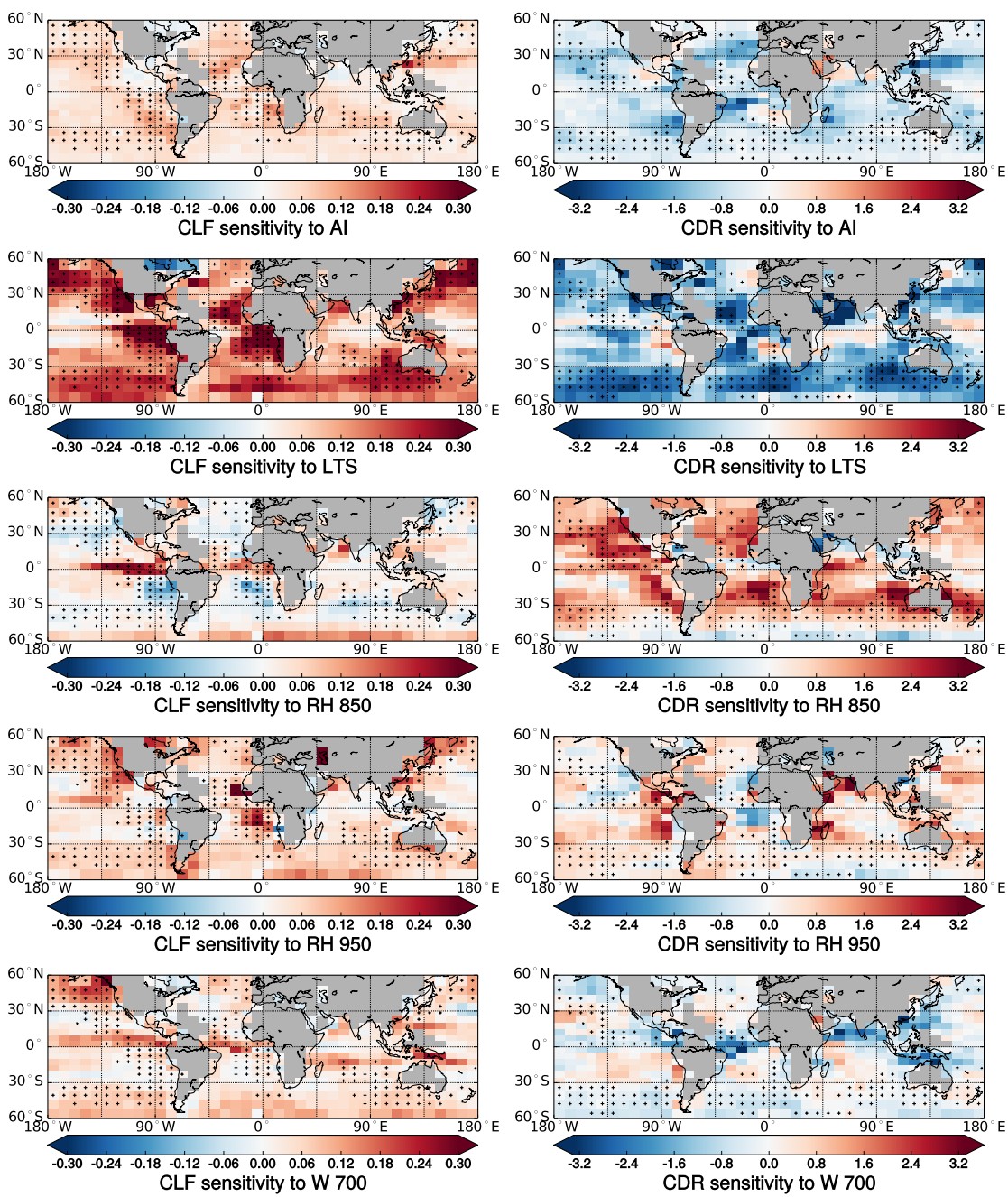

**Figure 6.** Global relative sensitivity patterns of selected CLF predictors in the left-hand column and CDR predictors in the right-hand column. Gray regions a) are over land or b) do not contain at least 2000 data points. Regions where the ANN test skill ($R^2$) is $> 0.5$ and rel. RMSE $<$ its global average are marked with a '+'.

RH at higher altitudes. CDR sensitivity to RH is stronger at higher pressure levels, where the cloud-tops are likely located. In addition to this, BLH is found to be a main determinant of LWP and COT. One should note though, that not all of the observed predictor-predictand sensitivities are necessarily a result of a direct physical relationship between the predictor and the predictand, but may in part be spurious due to cloud contamination of the satellite aerosol retrievals (Grandey et al., 2013), or due to the influence of confounding factors on both predictor and predictand (Gryspeerdt et al., 2016). For example, CLF and AI/AOD are both positively related to RH, potentially contributing to the observed positive CLF sensitivity to AI/AOD. Gryspeerdt et al. (2016) found that, by constraining potential aerosol-induced effects on CLF to situations where cloud droplet number concentration is simultaneously increased, the MODIS log(AOD)-CLF relationship is reduced by about 80 %. Issues of this kind are addressed here by including information on all relevant confounding factors directly in the ANN, and - for comparison measure - when all other inputs are held constant at their grid-cell specific average, the log(AI)-CLF relationship is on average about 40 % weaker than the originally observed log(AI)-CLF relationship. While the decrease in the sensitivity is not quite as strong as in Gryspeerdt et al. (2016), the results correspond well in the sense that bivariately determined aerosol-cloud sensitivities as in Quaas et al. (2008) are likely to overestimate aerosol indirect effects significantly. While the influence of confounding factors is limited by the multivariate approach, effects concerning data quality (e.g. cloud contamination) are not accounted for and need to be considered, especially when interpreting the CLF sensitivity to AI.

The ramifications of the interactions between aerosols and cloud occurrence and properties seem to be represented well in the ANN, following the general understanding of ACI. Specific regions of interest arise, such as the Northwest Atlantic with strong sensitivities to AI and regions that are affected by high dust loadings, with positive AI-CDR relationships and an above average positive AI-CLF sensitivity.

The results lead to the conclusion that on the system scale the aerosol may be viewed as a relevant determinant of marine liquid-water cloud fraction and microphysical properties, but only a secondary determinant for cloud optical thickness. On the scales considered here, lower tropospheric stability is the key controlling factor of cloud occurrence and droplet size, while boundary layer height controls the liquid water path and thus optical thickness of the cloud. The results confirm findings of previous studies that analyzed determinants of cloud properties in a more isolated manner (e.g. Klein and Hartmann, 1993; Johnson et al., 2004; Matsui et al., 2006). The results give confidence that the combination of observational and reanalysis data sets in a multivariate statistical approach is able to capture the natural variability of cloud occurrence and properties, and that meteorological and aerosol effects similar to those found in other studies can be identified in this system. In the future, a focussed, cloud-regime specific ANN approach similar to Gryspeerdt and Stier (2012) or Oreopoulos et al. (2016) could add to our system understanding. To address climate effects in a straight-forward manner, future research may also apply this study's approach to investigate the global determinants of cloud radiative effects.

## 5 Data availability

MODIS data used in this study were acquired as part of the NASA's Earth-Sun System Division and archived and distributed by the MODIS Adaptive Processing System (MODAPS). MODIS data were obtained from the Goddard Space Flight Cen-

ter (http://ladsweb.nascom.nasa.gov/data/search.html). ECMWF ERA-Interim data used in this study were obtained from the ECMWF data server (http://apps.ecmwf.int/datasets/data/interim-full-moda/levtype=sfc/).

*Author contributions.* J. Cermak had the initial idea and performed a precursor study. H. Andersen fully developed the method and the software, obtained and analyzed the data sets, conducted the original research and wrote the manuscript. H. Andersen, J. Cermak, J. Fuchs, R. Knutti and U. Lohmann contributed to study design and interpretation of findings.

*Competing interests.* The authors declare that they have no conflict of interest.

*Acknowledgements.* Funding for this study was provided by Deutsche Forschungsgemeinschaft (DFG) in the project GEOPAC (grant CE 163/5-1). Constructive and helpful comments from three anonymous reviewers are gratefully acknowledged.

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
