# Peer review of "Understanding the drivers of marine liquid-water cloud occurrence and properties with global observations using neural networks"

_Atmospheric Chemistry and Physics, 2017_

## Referee Comment (RC1) · Anonymous Referee #1 · 12 Apr 2017

I appreciate that the authors have attempted to diversify the ACI investigation field with the use of neural networks. It is often difficult with studies such as this that attempt a new analysis method to create a coherent message. However, I do not think this paper can be published in its current form. My overarching concern in this paper is that the authors do not articulate what the new thing is that they bring to the table besides the black box of a neural network.

I have grouped my concerns about this paper into the following categories:

Statistical evaluation: It is unclear to me why doing multiple neural networks on sub regions on monthly data tells us anything useful about what is going on. I need some sort of confidence that a high R2 model cannot be created by a large neural network

using a collection of meteorological predictors picked at random. Monthly data has the issue of being driven by the seasonal cycle, which will drive almost everything else, and making it regional will mean that the neural network doesn't need to tell us anything particularly meaningful about how the clouds are driven by their environment. The authors should consider using anomalies relative to the seasonal mean, or simply using annual means. Either of these options would be better than the approach taken in this paper. Admittedly the authors talk about this on page 2 line 25, but they don't provide any convincing proof that they haven't just created a regional seasonal cycle simulator.

On page 4/line 10 the authors note that they throw out models that have a low R2. I'm not sure why this is ok to do.

On page 6 I find this something of a straw man. A better test would be to compare a multiple linear regression of all the predictors to the ANN, as opposed to a regression on AOD alone. Or to compare the ANN trained using only AOD. I think that the paper would actually be vastly improved by just repeating the analysis with a multiple linear regression to demonstrate to skeptical readers why their paper brings anything new to the table as compared to the numerous previous papers that have looked at ACI and low cloud variability in the past.

Figure 5- If the error bars give the range in sensitivity does that mean that nothing except LTS and AOD have a robust relationship with cloud properties that holds outside of a few regions? Didn't we already know this very well from simple regression models that were easy to interpret (Klein & Hartmann, 1993; Nakajima, Higurashi, Kawamoto, & Penner, 2001)?

Choice of predictor/predictands: The choice of predictors by the authors is not appropriate for a paper in the last decade. Why have the authors chosen AOD to be a CCN proxy? AOD is not equivalent to CCN since it has a large contribution from larger, non-CCN relevant aerosols. Why don't the authors use AI, which is far more relevant and

typical of more recent studies (Patel, Quaas, & Kumar, 2017)? The authors acknowledge this, but then shrug this off because papers from almost a decade ago do it. In a similar vein, why do the authors use effective radius instead of CDNC? Effective radius for a fixed CCN increases with increasing LWC, making it sensitive to meteorological drivers. The authors do acknowledge this in page 10, section 25 noting that the interaction between inversion strength and effective radius is most likely driven by variations in LWC. This makes the interpretation of the CDR as a proxy for aerosol-cloud effects muddied. Further, the authors use LTS. Why not use EIS, which is used by every study investigating low cloud in the last decade (Myers & Norris, 2015; Qu, Hall, Klein, & Caldwell, 2014; Seethala, Norris, & Myers, 2015; Webb, Lambert, & Gregory, 2013)? Finally, I am concerned with the use of RH. Clouds and RH are a semi-equivalent quantity, which may just mean that they are comparing ECMWF-interim's cloud cover to MODIS, further aliasing in the seasonal cycle to their prediction model.

Writing: The writing is rushed and hard to follow. Clearly expressing why the methodology is valid is crucial for this study and as such the writing needs to be tightened up substantially to clarify their ideas.

Summary The authors articulate their guiding hypotheses, which I think is a good thing to do. I am not sure why (1) is a hypothesis. It seems to be more of a statement about neural networks and is worrisome since I am still concerned that the neural network is just looking at the seasonal cycle and is guaranteed to get a high R2. (2) is odd. Why would we have regional patterns? I could see it if this was a regime-dependent analysis (eg stratus vs convection), but the use of w and LTS as predictors in the neural network should mean that the authors can create a single neural network that effectively does this for them. Why is this not the case? What makes a specific lat-lon box a natural choice. (3) seems to imply that meteorology plays a secondary role to aerosols, which is not true. We don't expect aerosol to tell us where convection and stratus are, for instance.

Klein, S. A., & Hartmann, D. L. (1993).    The Seasonal Cycle of Low
Stratiform Clouds. Journal of Climate, 6(8), 1587-1606. doi:10.1175/1520-0442(1993)006<1587:tscols>2.0.co;2

Myers, T. A., & Norris, J. R. (2015). On the Relationships between Subtropical Clouds and Meteorology in Observations and CMIP3 and CMIP5 Models. Journal of Climate, 28(8), 2945-2967. doi:10.1175/JCLI-D-14-00475.1

Nakajima, T., Higurashi, A., Kawamoto, K., & Penner, J. E. (2001). A possible correlation between satellite-derived cloud and aerosol microphysical parameters. Geophysical Research Letters, 28(7), 1171-1174. doi:10.1029/2000GL012186

Patel, P. N., Quaas, J., & Kumar, R. (2017). A new statistical approach to improve the satellite-based estimation of the radiative forcing by aerosol–cloud interactions. Atmos. Chem. Phys., 17(5), 3687-3698. doi:10.5194/acp-17-3687-2017

Qu, X., Hall, A., Klein, S., & Caldwell, P. (2014). On the spread of changes in marine low cloud cover in climate model simulations of the 21st century. Climate Dynamics, 42(9-10), 2603-2626. doi:10.1007/s00382-013-1945-z

Seethala, C., Norris, J. R., & Myers, T. A. (2015). How Has Subtropical Stratocumulus and Associated Meteorology Changed since the 1980s? Journal of Climate, 28(21), 8396-8410. doi:10.1175/JCLI-D-15-0120.1

Webb, M. J., Lambert, F., & Gregory, J. M. (2013). Origins of differences in climate sensitivity, forcing and feedback in climate models. Climate Dynamics, 40(3-4), 677-707. doi:http://dx.doi.org/10.1007/s00382-012-1336-x

---

## Referee Comment (RC2) · Anonymous Referee #2 · 4 May 2017

This paper addresses a topic of significant current research, namely quantifying the effect of aerosols on cloud properties. The authors note the importance of local meteorology in determining the properties of clouds and that as meteorological factors are also correlated to aerosol properties, this can obscure the influence of aerosols on cloud properties. To explore the role of meteorology and aerosols, they make use of an artificial neural network (ANN) to examine the sensitivity of cloud properties to different predictors. Similar to previous studies, they show that meteorology is a strong control on the cloud properties, such that the cloud properties can be accurately predicted on a monthly timescale using reanalysis data and observed aerosol properties.

I think that this paper is a good addition to the literature on this topic, presenting a

new way to investigate the drivers of cloud properties. However, there are a couple of points, listed below, that I think should be clarified before publication. I particular, I think that using monthly data rather than daily/instantaneous data must be better justified. It would also make the paper stronger if the ANN method was compared to a more comparable statistical technique, such as a multiple linear regression across meteorological parameters. This might help to highlight the benefits of using an ANN, especially if it results in a different sensitivity of cloud properties to aerosol. Following these changes, I feel that this article would be suitable for publication in Atmospheric Chemistry and Physics.

Main points

While some previous studies have used monthly data for investigations into aerosol-cloud interactions, this disguises a lot of the variability in the cloud field and focuses on very large scale changes in cloud properties. The effect of seasonal variations can generate non-causal relationships between cloud properties and meteorological factors that might be accounted for if the study was done on a sub-seasonal scale using higher temporal resolution data. Can the authors explain why monthly data is used in this case and why daily data is unsuitable?

The use of an ANN seems to give a large improvement over just using AOD as a predictive variable for cloud properties. However, I am not sure this is a suitable comparison, as AOD is rarely assumed to be a good predictive variable for cloud properties on its own. As better comparison would be the predictive ability of (log) AOD on its own using a linear regression and from the ANN. Alternatively a comparison of a multiple linear regression and an ANN for predicting the cloud properties could show the added utility of using an ANN over existing methods. This might then highlight further useful properties of the ANN - for example, does it show a stronger (or weaker) sensitivity of cloud properties to aerosols when compared to current methods?

How do regional ANNs compare to a single global model? Presumably if enough meteorological parameters can be included, a single global model should be able to predict cloud properties everywhere. Requiring different models in different locations would then indicate that some meteorological parameter is missing from the ANN. A global pattern of the accuracy of the ANN might then give an indicator as to which parameters should be included. The ANN might be expected to differ as a function of cloud type, but perhaps a separate model for each cloud type (e.g. Gryspeerdt and Stier, 2012 or Oreopoulos et al., 2016) might be useful.

Minor points

P2L9: Perhaps only e.g. is necessary

P2L24: Why is the 2.1um effective radius used with the 3.7um LWP retrieval?

P2L29: Is the liquid fraction a suitable measure of cloud fraction, as it depends on the overlying ice cloud fraction? The authors could consider using cases where only liquid cloud exists in a gridbox, as this would remove this source of uncertainty.

P3L4: AOD is proportional to CCN (at least at some scales, see Andreae, 2009), it is just not a direct measurement (the same as with mass, as it also depends on aerosol optical properties)

P3L7: Many recent studies have used aerosol index (AOD times angstrom exponent) or a reanalysis aerosol parameter (e.g. Lebsock et al., 2008; McCoy et al., 2016). As these have been shown to more accurately predict cloud properties, they might further improve the skill of the ANN. Although MODIS AI is not necessarily accurate over land (Levy et al., 2013), it could be used over ocean in this study.

P3L13: It is definitely a good idea to investigate variables that have been previously used in aerosol-cloud studies. Koren et al., (2010) might also provide some useful guidance here. Although it was focussed on looking at convective clouds, some of the results (e.g. Figs. 8,9) might help decide which variables should be included in the ANN).

P4L33: Is there any significance behind using five hidden nodes?

P5L7: Are the sensitivities calculated using the local variation of meteorological values, or the same artificial values globally? If the relationship is non-linear and the mean values of the meteorological variables vary across the globe, this could strongly affect the calculated sensitivity.

P5L14: I am not sure I understand this sentence (which might explain my previous query?)

P5L20: If the other meteorological factors in the ANN are held constant, does this produce a different result for the simple sensitivity? (see main point)

P6L7: As I understand it previous work focusses on the sensitivity as this is related to the strength of the cloud response to aerosol. It is not often assumed that aerosols can explain much of the variability in cloud properties which might explain the low skill here.

P7L1: Perhaps another measure of skill might be useful in addition to the $R^2$? It could be argued that the skill in the shallow cumulus regions is quite good, in that the ANN (presumably) gets the cloud properties roughly right (the rms error might be small)?

P7L4: Does this removal of the poor skill models bias the results, perhaps as a function of meteorology (as would appear to be the case from the maps in Fig. 3)

P7L9: How does these sensitivities compare to previous results? Several studies have calculated AOD-CF or AOD-droplet number concentration sensitivities which could be compared here (e.g. Quaas et al (2008), Grandey et al. (2012), Gryspeerdt et al. (2016))

P12L3: Are the covariations really spurious? The argument here is not that the covariations don't exist, but that they are not representative of the causal relationship. I would suggest that if 'direct physical relationship' was replaced with 'causal relationship', this could instead mention the issue of confounding variables, similar to Gryspeerdt et al.,

(2016).

P12L4: To what extent has using RH in the ANN accounted for this effect?

References

Andreae, M. O. (2009), Correlation between cloud condensation nuclei concentration and aerosol optical thickness in remote and polluted regions, Atmos. Chem. Phys., 9(2), 543–556, doi:10.5194/acp-9-543-2009.

Gryspeerdt, E., and P. Stier (2012), Regime-based analysis of aerosol-cloud interactions, Geophysical Research Letters, 39, 21802, doi:10.1029/2012GL053221.

Gryspeerdt, E., J. Quaas, and N. Bellouin (2016), Constraining the aerosol influence on cloud fraction, J. Geophys. Res., 121(7), 3566–3583, doi:10.1002/2015JD023744.

Koren, I., G. Feingold, and L. A. Remer (2010), The invigoration of deep convective clouds over the Atlantic: aerosol effect, meteorology or retrieval artifact?, Atmos. Chem. Phys., 10(18), 8855–8872, doi:10.5194/acp-10-8855-2010.

Lebsock, M., G. Stephens, and C. Kummerow (2008), Multisensor satellite observations of aerosol effects on warm clouds, J. Geophys. Res., 113, D15205, doi:10.1029/2008JD009876.

Levy, R. C., S. Mattoo, L. A. Munchak, L. A. Remer, A. M. Sayer, F. Patadia, and N. C. Hsu (2013), The Collection 6 MODIS aerosol products over land and ocean, Atmos. Meas. Tech., 6(11), 2989–3034, doi:10.5194/amt-6-2989-2013.

McCoy, D. T., F. A.-M. Bender, J. K. C. Mohrmann, D. L. Hartmann, R. Wood, and D. P. Grosvenor (2017), The global aerosol-cloud first indirect effect estimated using MODIS, MERRA, and AeroCom, Journal of Geophysical Research: Atmospheres, 122, 1779–1796, doi:10.1002/2016JD026141.

Oreopoulos, L., N. Cho, D. Lee, and S. Kato (2016), Radiative effects of global MODIS cloud regimes, Journal of Geophysical Research: Atmospheres, n/a–n/a,

doi:10.1002/2015JD024502.

Quaas, J., O. Boucher, N. Bellouin, and S. Kinne (2008), Satellite-based estimate of the direct and indirect aerosol climate forcing, J. Geophys. Res., 113, 05204, doi:10.1029/2007JD008962.

---

## Author Comment (AC1) · 23 May 2017

**Understanding the drivers of marine liquid-water cloud occurrence and properties with global observations using neural networks**

**— RESPONSE TO REFEREE 1 —**

contact: hendrik.andersen@kit.edu

We would like to thank referee 1 for her/his review of the manuscript and her/his constructive criticism. Comments by the referee are colored in blue, our replies are colored in black.

We have thoroughly considered and discussed your input and after careful analysis of each review point concur with you that we have indeed not sufficiently 'articulate[d] what the new thing is that [we] bring to the table', as you state as your 'overarching concern'. The work in this manuscript has a history of several years, over which we have discussed ideas and results with peers and internally many times, so that in writing the manuscript we may have taken several points for granted that are in fact new to a reader confronted with the study for the first time. In this spirit, we have now attempted, guided by your suggestions, to more carefully explain the whats, hows and whys of our research, as well as what is new, and what is not.

I appreciate that the authors have attempted to diversify the ACI investigation field with the use of neural networks. It is often difficult with studies such as this that attempt a new analysis method to create a coherent message. However, I do not think this paper can be published in its current form. My overarching concern in this paper is that the authors do not articulate what the new thing is that they bring to the table besides the black box of a neural network.

General response: See above. Figure 1 included in this document is intended to illustrate the concept of our study schematically: Frequently, aerosol-cloud interactions are studied in a rather isolated manner (in red). At the same time, it is commonly acknowledged that the influence of aerosols is modulated by many environmental factors. With this study, we aim at analyzing the aerosol-cloud-climate system in its entirety. This includes all variations in the environmental

conditions, including the seasonal cycle (and its variability) of clouds and meteorology. Our first aim therefore is to find a way to statistically capture this system as completely as possible, including seasonality. Then, in a second step, we focus on and try to separate the effects of aerosols on cloud occurrence and properties from everything else. Our work is not intended to refute previous work done in this field. On the contrary: We would argue that most of the results presented within the study confirm many known aspects of the aerosol-cloud-climate system. But the fact that we were able to find these relationships in a statistical approach considering much more than only aerosol and cloud properties adds an additional line of independent evidence that strengthens the confidence in the existing system understanding. However, this is achieved without isolating specific processes of interest but rather by viewing the system in its entirety. Accordingly, these are the main new things we 'bring to the table': Confidence that the observation data sets considered in a multivariate statistical approach capture the natural variability, and that aerosol effects similar to those found in other studies can be identified in this system. No more, no less.

[Figure]

Figure 1: A schematic illustration of the concept of this study (ACS: aerosol-cloud sensitivity).

I have grouped my concerns about this paper into the following categories:

**Statistical evaluation**

1) It is unclear to me why doing multiple neural networks on sub regions on monthly data tells us anything useful about what is going on. I need some sort of confidence that a high R2 model cannot be created by a large neural network using a collection of meteorological predictors picked at random. Monthly data has the issue of being driven by the seasonal cycle, which will drive almost everything else, and making it regional will mean that the neural network doesn't need to tell us anything particularly meaningful about how the clouds are driven by their environment. The authors should consider using anomalies relative to the seasonal mean, or simply using annual means. Either of these options would be better than the approach taken in this paper. Admittedly the authors talk about this on page 2 line 25, but they don't provide any convincing proof that they haven't just created a regional seasonal cycle simulator.

This is related to what we argue above: We intend to model liquid-water clouds including their seasonal cycle by using information on aerosol loading and a set of meteorological drivers that were identified as main drivers of liquid-water clouds after careful study of current literature. One could probably create a relatively high $R^2$ model with a very large array of randomly selected predictors due to spurious covariation of seasonal cycles between predictors and predictands. However, in this study, we avoid this by capturing the aerosol-cloud-climate system with a small number of the known main drivers of cloud occurrence and properties. Within this modeled system we then try to understand the effects of each driver and its regional patterns. We argue that regionally specific neural networks are needed to capture the regional variability of liquid-water clouds. Regional patterns exist due to regional differences in

cloud type, aerosol composition, meteorology and the respective seasonal cycles.

2) On page 4/line 10 the authors note that they throw out models that have a low R2. I'm not sure why this is ok to do.

We have identified $R^2$ and the root mean square error relative to the mean as good indicators for model skill. We are interested in understanding predictor-predictand relationships by analyzing their respective sensitivities, however, we choose to trust only models that can adequately represent the observed cloud patterns. We prefer to err on the side of caution to avoid reaching conclusions based on inadequate statistical relationships; thus we exclude models that in our opinion are not capable of representing the system well enough. We are open to other ideas regarding alternative ways to ensure adequate model skill.

We probably did not communicate the intention of this figure with sufficient clarity: This figure is intended to show how well a combination of aerosol and meteorological conditions can explain the variance of cloud properties (multivariate statistics) as opposed to a simple bivariate approach. We have added results of a multiple linear regression using all the ANN predictors to the figure (2). The comparison of the results of the multiple linear regression and the ANN

[Figure]

Figure 2: Predictand correlation with ANN (multivariate) test output, multiple linear regression (multivariate) and log(AI) (bivariate). The median is represented by the black horizontal line, framed by the interquartile range (boxes), whiskers expand the boxes by 1.5 interquartile ranges.

suggest that the ANN is an appropriate method to be used in this context.

Neural networks were our statistical method of choice, as they have the advantage of not being reliant on statistical assumptions on predictor and predictand distributions and they are capable of modeling nonlinear relationships. That being said, we agree that other multivariate methods (e.g. multiple linear regression) could also have been used.

4) Figure 5- If the error bars give the range in sensitivity does that mean that nothing except LTS and AOD have a robust relationship with cloud properties that holds outside of a few regions? Didn't we already know this very well from

simple regression models that were easy to interpret (Klein & Hartmann, 1993; Nakajima, Higurashi, Kawamoto, & Penner, 2001)?

We agree with the referee that many of the results of this study confirm what previous studies have already shown. Since we have reached these conclusions using a different methodology, we add another line of evidence. The lack of other relevant relationships would not have been obvious without such an analysis. In our opinion the value of our study is that the results were produced by looking at the entire system at once rather than at isolated relationships. Using this method, we can compare the relevance of each predictor to each predictand including spatial patterns.

5) Choice of predictor/predictands: The choice of predictors by the authors is not appropriate for a paper in the last decade. Why have the authors chosen AOD to be a CCN proxy? AOD is not equivalent to CCN since it has a large contribution from larger, non-CCN relevant aerosols. Why don't the authors use AI, which is far more relevant and typical of more recent studies (Patel, Quaas, & Kumar, 2017)? The authors acknowledge this, but then shrug this off because papers from almost a decade ago do it. In a similar vein, why do the authors use effective radius instead of CDNC? Effective radius for a fixed CCN increases with increasing LWC, making it sensitive to meteorological drivers. The authors do acknowledge this in page 10, section 25 noting that the interaction between inversion strength and effective radius is most likely driven by variations in LWC. This makes the interpretation of the CDR as a proxy for aerosol-cloud effects muddied. Further, the authors use LTS. Why not use EIS, which is used by every study investigating low cloud in the last decade (Myers & Norris, 2015; Qu, Hall, Klein, & Caldwell, 2014; Seethala, Norris, & Myers, 2015; Webb, Lambert, & Gregory, 2013)? Finally, I am concerned with the use

139 of RH. Clouds and RH are a semi-equivalent quantity, which may just mean that

140 they are comparing ECMWF-interim's cloud cover to MODIS, further aliasing

141 in the seasonal cycle to their prediction model.

142 AOD vs. AI: For this study, we used the newest version of MODIS products

143 available, collection 6 (C6). In C6, the MODIS Ångström exponent (needed

144 for the computation of the aerosol index as it is the product of AOD and the

145 Ångström exponent) has been discontinued in level 3 (L3) data (p. 3018 Levy

146 et al., 2013). We believe that for this and for other reasons, other recent studies

147 also use the AOD as a proxy for CCN (see: Gryspeerdt and Stier, 2012; Tang

148 et al., 2014; Chakraborty et al., 2016; Stathopoulos et al., 2017; Patel et al.,

149 2017). We agree with referee 1 though, that the aerosol index is an appropriate

150 measure for CCN and have chosen to use it in the ANN. The following figures

151 3 and 4 are the new results of the ANN when using AI instead of AOD. The

152 spatial patterns in the ANN skill, as well as the mean global sensitivities are

153 nearly identical (compare with figures 3 and 5 in the original ACPD manuscript).

[Figure]

Figure 3: Global patterns of ANN skill as in the manuscript; AI has been used instead of AOD.

[Figure]

Figure 4: Global mean relative sensitivities as in the manuscript; AI has been used instead of AOD.

Small differences can be observed in the regional patterns of ANN sensitivities (fig. 5 on the following page). The CLF sensitivity to AI is higher in the Southeast Atlantic than its sensitivity to AOD in that specific region. The Southeast Atlantic is of course dominated by biomass burning aerosol, which are mostly in the fine mode and thus feature a relatively larger AI than AOD. The sensitivity of CDR to AI differs from its sensitivity to AOD in regions that are dominated by desert dust. Dust is relatively coarse, so that the AI would be underproportional to the AOD in these regions which might explain the differences between the sensitivities of the two.

[Figure]

Figure 5: Difference in sensitivities of CLF and CDR to AI (left-hand column) vs. AOD (right-hand column).

CDR vs. CDNC: We agree with the referee that CDNC is a better quantity for the direct analysis of the first aerosol indirect effect, however, its retrieval from satellite is quite problematic, as the retrieval of CDNC requires additional assumptions on the cloud water profile. The commonly-applied adiabatic assumption might be a good proxy for many regions and cloud types (i.e. stratocumulus clouds), however, we are investigating all liquid-water clouds on a global scale. Bennartz and Rausch (2017) showed that the uncertainties in the CDNC retrievals are significantly increased in non-stratocumulus regions. As we are investigating global patterns for various liquid-water cloud types, we came to the conclusion that the uncertainty related to the CDNC retrievals outweighs the theoretical advantages of using CDNC rather than CDR.

LTS vs. EIS: We do not see a specific advantage of using EIS over LTS, as e.g. Lacagnina and Selten (2013) found that for the Californian stratus, LTS is a better predictor than EIS. Some other recent studies that use LTS are e.g. George and Wood (2010); Chen et al. (2014); Gryspeerdt et al. (2014, 2016); Painemal et al. (2014a,b); Adebiyi et al. (2015); Adebiyi and Zuidema (2016); Coopman

et al. (2016); Eastman et al. (2016); Ghan et al. (2016). That being said, we would agree that EIS is an appropriate alternative measure for large-scale thermodynamics.

RH: As pointed out above, our intention is to capture the entire aerosol-cloud-climate system and in our opinion, relative humidity has a key role within this system. Thus, the inclusion of RH in the model was a necessity.

**Writing:**

The writing is rushed and hard to follow. Clearly expressing why the methodology is valid is crucial for this study and as such the writing needs to be tightened up substantially to clarify their ideas.

See our comment at the beginning of this letter. We will attempt to describe the reason for the methodology, the hypotheses and the relevance of our work more clearly in the revised manuscript.

**Summary:**

The authors articulate their guiding hypotheses, which I think is a good thing to do. I am not sure why (1) is a hypothesis. It seems to be more of a statement about neural networks and is worrisome since I am still concerned that the neural network is just looking at the seasonal cycle and is guaranteed to get a high R2. (2) is odd. Why would we have regional patterns? I could see it if this was a regime-dependent analysis (eg stratus vs convection), but the use of w and LTS as predictors in the neural network should mean that the authors can create a single neural network that effectively does this for them. Why is this not the case? What makes a specific lat-lon box a natural choice.

(3) seems to imply that meteorology plays a secondary role to aerosols, which is not true. We don't expect aerosol to tell us where convection and stratus are, for instance.

1) Neural networks have not been used in this context before, so their capabilities in this context were not quite clear. This is also the case for the separation of aerosol and meteorological effects.

2) While this study does not contrast e.g. stratus vs. convection, we analyze all liquid-water clouds globally. It is clear that these feature different cloud types in different regions and that different processes drive these different clouds. This is shown in figure 6. Regional patterns in aerosol-cloud sensitivity exist. They have been shown to be dependent on meteorology and aerosol species composition (e.g. Andersen et al., 2016). If we created a single neural network, all of the regional characteristics and regionally specific sensitivities (c.f. figure 6) would be blurred or missed completely.

3) Our third hypothesis is certainly not intended to imply that meteorology plays a secondary role to aerosols. We will change the wording for clarity in the revised manuscript.

**References**

Adebiyi, A. A. and Zuidema, P. (2016). The role of the southern African easterly jet in modifying the southeast Atlantic aerosol and cloud environments. *Quarterly Journal of the Royal Meteorological Society*, 142(697):1574–1589.

Adebiyi, A. A., Zuidema, P., and Abel, S. J. (2015). The Convolution of Dynamics and Moisture with the Presence of Shortwave Absorbing Aerosols over the Southeast Atlantic. *Journal of Climate*, 28(5):1997–2024.

Andersen, H., Cermak, J., Fuchs, J., and Schwarz, K. (2016). Global observations of cloud-sensitive aerosol loadings in low-level marine clouds. *Journal of Geophysical Research: Atmospheres*, 121(21):12936–12946.

Bennartz, R. and Rausch, J. (2017). Global and regional estimates of warm cloud droplet number concentration based on 13 years of aqua-modis observations. *Atmospheric Chemistry and Physics Discussions*, 2017:1–32.

Chakraborty, S., Fu, R., Massie, S. T., and Stephens, G. (2016). Relative influence of meteorological conditions and aerosols on the lifetime of mesoscale convective systems. *Proceedings of the National Academy of Sciences of the United States of America*, 113(27):7426–7431.

Chen, Y.-C., Christensen, M. W., Stephens, G. L., and Seinfeld, J. H. (2014). Satellite-based estimate of global aerosol-cloud radiative forcing by marine warm clouds. *Nature Geoscience*, 7(9):643–646.

Coopman, Q., Garrett, T. J., Riedi, J., Eckhardt, S., and Stohl, A. (2016). Effects of long-range aerosol transport on the microphysical properties of low-level liquid clouds in the Arctic. *Atmospheric Chemistry and Physics*, 16(7):4661–4674.

Eastman, R., Wood, R., Eastman, R., and Wood, R. (2016). Factors Controlling Low-Cloud Evolution over the Eastern Subtropical Oceans: A Lagrangian Perspective Using the A-Train Satellites. *Journal of the Atmospheric Sciences*, 73(1):331–351.

George, R. C. and Wood, R. (2010). Subseasonal variability of low cloud radiative properties over the southeast Pacific Ocean. *Atmospheric Chemistry and Physics*, 10(8):4047–4063.

Ghan, S., Wang, M., Zhang, S., Ferrachat, S., Gettelman, A., Griesfeller, J., Kipling, Z., Lohmann, U., Morrison, H., Neubauer, D., Partridge, D. G.,

Stier, P., Takemura, T., Wang, H., and Zhang, K. (2016). Challenges in constraining anthropogenic aerosol effects on cloud radiative forcing using present-day spatiotemporal variability. *Proceedings of the National Academy of Sciences of the United States of America*, 113(21):5804–5811.

Gryspeerdt, E., Quaas, J., and Bellouin, N. (2016). Constraining the aerosol influence on cloud fraction. *Journal of Geophysical Research: Atmospheres*, 121:3566–3583.

Gryspeerdt, E. and Stier, P. (2012). Regime-based analysis of aerosol-cloud interactions. *Geophysical Research Letters*, 39(21):L21802.

Gryspeerdt, E., Stier, P., and Partridge, D. G. (2014). Satellite observations of cloud regime development: the role of aerosol processes. *Atmospheric Chemistry and Physics*, 14(3):1141–1158.

Lacagnina, C. and Selten, F. (2013). A novel diagnostic technique to investigate cloud-controlling factors. *Journal of Geophysical Research: Atmospheres*, 118(12):5979–5991.

Levy, R. C., Mattoo, S., Munchak, L. A., Remer, L. A., Sayer, A. M., Patadia, F., and Hsu, N. C. (2013). The Collection 6 MODIS aerosol products over land and ocean. *Atmospheric Measurement Techniques*, 6(11):2989–3034.

Painemal, D., Kato, S., and Minnis, P. (2014a). Boundary layer regulation in the southeast Atlantic cloud microphysics during the biomass burning season as seen by the A-train satellite constellation. *Journal of Geophysical Research: Atmospheres*, 119(19):11288–11302.

Painemal, D., Road, N., and Stop, M. (2014b). Mean Structure and diurnal cycle of Southeast Atlantic boundary layer clouds: Insights from satellite observations and multiscale modeling framework simulations. *journal of climate*, pages 1–51.

Patel, P. N., Quaas, J., and Kumar, R. (2017). A new statistical approach to improve the satellite based estimation of the radiative forcing by aerosol–cloud interactions. *Atmospheric Chemistry and Physics Discussions*, 17:3687–3698.

Stathopoulos, S., Georgoulias, A., and Kourtidis, K. (2017). Space-borne observations of aerosol - cloud relations for cloud systems of different heights. *Atmospheric Research*, 183:191–201.

Tang, J., Wang, P., Mickley, L. J., Xia, X., Liao, H., Yue, X., Sun, L., and Xia, J. (2014). Positive relationship between liquid cloud droplet effective radius and aerosol optical depth over Eastern China from satellite data. *Atmospheric Environment*, 84:244–253.

---

## Author Comment (AC2) · 23 May 2017

Understanding the drivers of marine liquid-water

cloud occurrence and properties with global

observations using neural networks

— RESPONSE TO REFEREE 2 —

contact: hendrik.andersen@kit.edu

We would like to thank referee 2 for her/his review of the manuscript and her/his constructive criticism. Comments by the referee are colored in blue, our replies are colored in black.

This paper addresses a topic of significant current research, namely quantifying the effect of aerosols on cloud properties. The authors note the importance of local meteorology in determining the properties of clouds and that as meteorological factors are also correlated to aerosol properties, this can obscure the influence of aerosols on cloud properties. To explore the role of meteorology and aerosols, they make use of an artificial neural network (ANN) to examine the sensitivity of cloud properties to different predictors. Similar to previous studies, they show that meteorology is a strong control on the cloud properties, such that the cloud properties can be accurately predicted on a monthly timescale using reanalysis data and observed aerosol properties.

I think that this paper is a good addition to the literature on this topic, presenting a new way to investigate the drivers of cloud properties. However, there are a couple of points, listed below, that I think should be clarified before publication. I particular, I think that using monthly data rather than daily/instantaneous data must be better justified. It would also make the paper stronger if the ANN method was compared to a more comparable statistical technique, such as a multiple linear regression across meteorological parameters. This might help to highlight the benefits of using an ANN, especially if it results in a different sensitivity of cloud properties to aerosol. Following these changes, I feel that this article would be suitable for publication in Atmospheric Chemistry and Physics.

We respond to each point individually below.

**Main points**

1) While some previous studies have used monthly data for investigations into aerosol-cloud interactions, this disguises a lot of the variability in the cloud field and focuses on very large scale changes in cloud properties. The effect of seasonal variations can generate non-causal relationships between cloud properties and meteorological factors that might be accounted for if the study was done on a sub-seasonal scale using higher temporal resolution data. Can the authors explain why monthly data is used in this case and why daily data is unsuitable?

With this study, we specifically aim at analyzing the aerosol-cloud-climate system at a very large scale ('system scale'). The monthly time scale is used here, as a) this enables a focus on the large-scale patterns and relationships and b) GCM output is also at a monthly time scale, so that future comparisons between our observationally-based results and GCMs can be conducted. We acknowledge the 'non-causal relationship' argument by referee 2 by using only a very limited number predictors in ANNs that have previously been shown to be the main drivers of liquid-water clouds. The results of the ANNs are physically plausible (signs, magnitudes and regional patterns of the sensitivities) and give another line of independent evidence that strengthens the confidence in our current system understanding. That being said, we cannot exclude the possibility that some of the observed relationships might be in part non-causal (which is true for other averaging time scales as well).

2) The use of an ANN seems to give a large improvement over just using AOD as a predictive variable for cloud properties. However, I am not sure this is a suitable comparison, as AOD is rarely assumed to be a good predictive variable for cloud properties on its own. As better comparison would be the predictive

ability of (log) AOD on its own using a linear regression and from the ANN. Alternatively a comparison of a multiple linear regression and an ANN for predicting the cloud properties could show the added utility of using an ANN over existing methods. This might then highlight further useful properties of the ANN - for example, does it show a stronger (or weaker) sensitivity of cloud properties to aerosols when compared to current methods?

We probably did not communicate the intention of this figure with sufficient clarity: This figure was simply intended to show how well a combination of aerosol and meteorological conditions can explain the variance of cloud properties (multi-variate statistics) as opposed to a simple bivariate approach. We have added results of a multiple linear regression using all the ANN predictors to the figure as suggested to illustrate the skill of the ANN vs. another multivariate method. The comparison of the results of the multiple linear regression and the ANN suggests that the ANN is an appropriate method to be used in this context. As suggested, we have switched from using the AOD to the AI and used log(AI) for this figure.

[Figure]

Figure 1: Predictand correlation with ANN (multivariate) test output, multiple linear regression (multivariate) and log(AI) (bivariate).

3) How do regional ANNs compare to a single global model? Presumably if enough meteorological parameters can be included, a single global model should be able to predict cloud properties everywhere. Requiring different models in different locations would then indicate that some meteorological parameter is missing from the ANN. A global pattern of the accuracy of the ANN might then give an indicator as to which parameters should be included. The ANN might be expected to differ as a function of cloud type, but perhaps a separate model for each cloud type (e.g. Gryspeerdt and Stier, 2012 or Oreopoulos et al., 2016) might be useful.

If one trains a single global model to predict CLF, using the same predictors and model setup as for the regional ANNs, it cannot predict CLF as well as most regional ANNs ($R^2$ of global model $\approx$ 0.45; median of regional ANNs $>$ 0.60). While adding additional predictors to the global ANN could still improve the skill of the model, it is unrealistic to think that a single model could represent clouds as well as regional models can (it would also increase the probability of non-causal relationships). Regional ANNs are superior, as they are able to reproduce the regionally varying predictor-predictand relationships (c.f. fig. 6 in the manuscript). These regional differences would be blurred or missed completely when using a single global ANN. Regional ANNs also have the advantage that knowledge on typical regional characteristics (e.g. aerosol species composition) can be included in the interpretation of the results (as in Andersen et al., 2016). That being said, cloud type-specific ANNs seem to be an interesting idea for future work.

**Minor points**

We agree and have changed the manuscript accordingly.

102

P2L24: Why is the 2.1um effective radius used with the 3.7um LWP retrieval?

We have changed the cloud products used (see our response below).

P2L29: Is the liquid fraction a suitable measure of cloud fraction, as it depends on the overlying ice cloud fraction? The authors could consider using cases where only liquid cloud exists in a gridbox, as this would remove this source of uncertainty.

After internal and peer discussions, we have decided to run the ANN with monthly means of single layer clouds only. While the results are nearly identical, the argument is valid, so that we only use single layer cloud products the current version of the manuscript.

P3L4: AOD is proportional to CCN (at least at some scales, see Andreae, 2009), it is just not a direct measurement (the same as with mass, as it also depends on aerosol optical properties)

Yes, we agree. We have corrected this in the revised manuscript.

P3L7: Many recent studies have used aerosol index (AOD times angstrom exponent) or a reanalysis aerosol parameter (e.g. Lebsock et al., 2008; McCoy et al., 2016). As these have been shown to more accurately predict cloud properties, they might further improve the skill of the ANN. Although MODIS AI is not necessarily accurate over land (Levy et al., 2013), it could be used over ocean in this study.

For this study, we used the newest version of MODIS products available, collection 6 (C6). In C6, the MODIS Ångström exponent (needed for the computation of the aerosol index as it is the product of AOD and the Ångström exponent) has

been discontinued in level 3 (L3) data (p. 3018 Levy et al., 2013). We believe
that for this and for other reasons, other recent studies also use the AOD as a
proxy for CCN (e.g. Chakraborty et al., 2016; Stathopoulos et al., 2017; Patel
et al., 2017). We agree with the referee though that the aerosol index might
be a more appropriate measure for CCN and have thus chosen to compute the
Ångström exponent (550 and 867nm) ourselves to use aerosol index instead of
AOD in the ANN. The following figures 2 and 3 are the new results of the ANN
when using AI instead of AOD. The spatial patterns in ANN skill, as well as the
mean global sensitivities are nearly identical (cf. figures 3 and 5 in the original
ACPD manuscript).

[Figure]

Figure 2: Global patterns of ANN skill as in the manuscript; AI has been used
instead of AOD.

Small differences can be observed in the regional patterns of ANN sensitivi-
ties (fig. 4) to AI vs. AOD. The CLF sensitivity to AI is higher in the Southeast
Atlantic than its sensitivity to AOD in that specific region. The Southeast At-
lantic is of course dominated by biomass-burning aerosols, which are mostly fine
mode and thus feature a relatively larger AI than AOD. The sensitivity of CDR

[Figure]

Figure 3: Global mean relative sensitivities as in the manuscript; AI has been used instead of AOD.

to AI differs from its sensitivity to AOD in regions that are dominated by desert dust. Dust is relatively coarse, so that the AI would be disproportionally lower than the AOD in these regions, which might explain the differences between the sensitivities of the two.

[Figure]

Figure 4: Differences in sensitivities of CLF and CDR to AI (left-hand column) vs. AOD (right-hand column).

P3L13: It is definitely a good idea to investigate variables that have been previously used in aerosol-cloud studies. Koren et al., (2010) might also provide some useful guidance here. Although it was focussed on looking at convective clouds, some of the results (e.g. Figs. 8,9) might help decide which variables should be included in the ANN).

We agree that additional variables (e.g. geopotential height, horizontal winds) might improve the ANN performance in some regions. Our goal in predictor selection was to minimize the number of predictors to a few key variables, in order to prevent covariation between the predictors. Also, additional predictors increase the probability of highlighting non-causal relationships.

P4L33: Is there any significance behind using five hidden nodes?

After thorough testing, five hidden nodes appeared to be a good global number. In general, the optimum number of nodes is dependent on the problem at hand. The number of nodes needed is connected to the complexity of the relationships, the amount of noise in the data and the amount of training data available. Too

many nodes can lead to overfitting and poor generalization, whereas the ANN may not converge to a global minimum when too few nodes are used (Gardner and Dorling, 1998). We found that while regional ANNs may differ, five nodes where a reasonable choice, as additional nodes typically only marginally, if at all, increased model skill. To illustrate this, figure 5 is an example of the effect of the number of hidden nodes on ANN skill in the Southeast Atlantic region. This figure is obviously not the basis for our decision to use 5 nodes, but is intended to illustrate a typical example for the dependence of a regional ANN skill on the number of hidden nodes.

[Figure]

Figure 5: Example (Southeast Atlantic) for the effect of the number of hidden nodes in the ANN.

P5L7: Are the sensitivities calculated using the local variation of meteorological values, or the same artificial values globally? If the relationship is non-linear and the mean values of the meteorological variables vary across the globe, this could strongly affect the calculated sensitivity.

This sentence was intended to describe how sensitivities can generally be computed with an ANN. In the text passage further down (P5L14), we describe

how sensitivities are computed in this study. To answer your question: Yes, the sensitivities are calculated using the local variation of meteorological values ('grid cell specific mean values'). In the revised version of the manuscript, we will attempt to describe both text passages more clearly.

P5L14: I am not sure I understand this sentence (which might explain my previous query?)

We compute ANN-predicted outputs for two groups of input data:

- All grid-cell specific retrievals of a specific predictor smaller than its 25th percentile.

- All grid-cell specific retrievals of a specific predictor greater than its 75th percentile.

In all cases, all other predictors are held constant at their grid-cell specific mean values. We then compute the average of both groups of ANN-predicted outputs. The difference between the two averages is defined as the sensitivity of the predictand to the specific predictor that was varied. We will try to more clearly describe this in the revised version of the manuscript.

P5L20: If the other meteorological factors in the ANN are held constant, does this produce a different result for the simple sensitivity? (see main point)

We have tested this for the sensitivity of CLF to AI. As above, we have also used data from the the Southeast Atlantic for this example. We found that the sensitivity (linear slope of AI-CLF relationship) of CLF to AI is $\approx 40\,\%$ lower in the ANN than in the observations. This is, of course, because in the sensitivity of the ANN, the other predictors are held constant, constraining their effect on CLF. This corresponds rather well to Gryspeerdt et al. (2016) who found that the sensitivity of CLF to AOD is reduced even further (80 %) when including

information on CDNC along the causal pathway of the AOD-CLF relationship.

P6L7: As I understand it previous work focusses on the sensitivity as this is related to the strength of the cloud response to aerosol. It is not often assumed that aerosols can explain much of the variability in cloud properties which might explain the low skill here.

Yes, we agree. This figure is not intended to illustrate sensitivities, but that we are in a space of large uncertainty when we derive sensitivities using bivariate methods. Using a multivariate approach (also the case for multiple regression, as outlined above) we are capturing more of the aerosol-cloud climate system. The derived sensitivities might thus be more reliable.

P7L1: Perhaps another measure of skill might be useful in addition to the $R^2$? It could be argued that the skill in the shallow cumulus regions is quite good, in that the ANN (presumably) gets the cloud properties roughly right (the rms error might be small)?

Yes, indeed, we also looked at the relative RMSE. Actually, the a combination of relative RMSE and $R^2$ thresholds (P7L4) are used to select the regions that are used for the computation of sensitivities (marked with a '+' in the maps). The relative RMSE and $R^2$ are basically invertly related.

P7L4: Does this removal of the poor skill models bias the results, perhaps as a function of meteorology (as would appear to be the case from the maps in Fig. 3)

The computed sensitivities are only valid for the regions and are not intended to be "global" in that sense.

P7L9: How does these sensitivities compare to previous results? Several studies have calculated AOD-CF or AOD-droplet number concentration sensitivities which could be compared here (e.g. Quaas et al (2008), Grandey et al. (2012), Gryspeerdt et al. (2016))

We compute the sensitivity a slightly different way, so a straight-forward comparison is not possible. However, in a similar way that Gryspeerdt et al. (2016) constrain the aerosol-CLF relationship with CDNC, the ANN constrains the aerosol-cloud relationships by meteorology. In the updated version of the manuscript, we will include comparisons to sensitivities found by other recent studies.

P12L3: Are the covariations really spurious? The argument here is not that the covariations don't exist, but that they are not representative of the causal relationship. I would suggest that if 'direct physical relationship' was replaced with 'causal relationship', this could instead mention the issue of confounding variables, similar to Gryspeerdt et al.,(2016).

We will restructure this text passage in the updated version of the manuscript.

P12L4: To what extent has using RH in the ANN accounted for this effect?

As shown in figure 6 within this document, the sensitivity of CLF to AI is weakened in the ANN, probably due to the meteorological constrains of the model. These are hard to track down to a single predictor, though (e.g. RH). It is likely that the main confounding factor for this relationship is RH and that most of the change in AI-CLF sensitivity is due to constraining RH.

**References**

Andersen, H., Cermak, J., Fuchs, J., and Schwarz, K. (2016). Global observations of cloud-sensitive aerosol loadings in low-level marine clouds. *Journal of Geophysical Research: Atmospheres*, 121(21):12936–12946.

Chakraborty, S., Fu, R., Massie, S. T., and Stephens, G. (2016). Relative influence of meteorological conditions and aerosols on the lifetime of mesoscale convective systems. *Proceedings of the National Academy of Sciences of the United States of America*, 113(27):7426–7431.

Gardner, M. and Dorling, S. (1998). Artificial neural networks (the multilayer perceptron) – a review of applications in the atmospheric sciences. *Atmospheric Environment*, 32(14):2627–2636.

Gryspeerdt, E., Quaas, J., and Bellouin, N. (2016). Constraining the aerosol influence on cloud fraction. *Journal of Geophysical Research: Atmospheres*, 121:3566–3583.

Levy, R. C., Mattoo, S., Munchak, L. A., Remer, L. A., Sayer, A. M., Patadia, F., and Hsu, N. C. (2013). The Collection 6 MODIS aerosol products over land and ocean. *Atmospheric Measurement Techniques*, 6(11):2989–3034.

Patel, P. N., Quaas, J., and Kumar, R. (2017). A new statistical approach to improve the satellite based estimation of the radiative forcing by aerosol–cloud interactions. *Atmospheric Chemistry and Physics Discussions*, 17:3687–3698.

Stathopoulos, S., Georgoulias, A., and Kourtidis, K. (2017). Space-borne observations of aerosol - cloud relations for cloud systems of different heights. *Atmospheric Research*, 183:191–201.

---

## Referee Comment (RC3) · Anonymous Referee #3 · 26 May 2017

This paper pursues a promising approach to study the sensitivity of marine liquid-water cloud properties on a set of meteorological and aerosol predictors, using an artificial neural network approach. It steers clear of correlative approaches for studying aerosol-cloud interactions and instead considers the meteorological context, segregated by region / meteorological regime. In essence, this amounts to a multi-variate analysis based on an optimal combination of satellite and re-analysis data. The paper is very well written, clearly represents new ideas, and has the potential to lead to major improvements in our assessment of ACI, regionally and globally. It is rare to see such a high-quality paper. I only have minor comments, which don't necessarily have to be addressed in this manuscript, but could be considered in future work. The most important ones are probably #1 regarding scale, and regarding the quality (reliability) of the data. Also, follow-up papers might consider using the co-sensitivity of some predictors (details below).

In a separate comment to the editor, I recommended that the paper be highlighted because it seems highly innovative in its approach and deviates from the traditional correlative aerosol-cloud interaction studies. I believe that it has potential to change the direction of this field of research.

General comments:

p5,L18: In the spirit of the McComiskey and Feingold ACI papers, it would have been interesting to also consider the impact of scale on ACI relationships. Here, one specific scale has been used (dictated by the analysis grid) - but it may not be straightforward to generalize these relationships.

p6,L4: "skill of simple correlation between AOD & cloud properties": It is a bit unclear, which "simple correlations" specifically have been used for this study. This statement calls for elaboration. The statement on p6,L6/7 shows the intent - the "simple correlations" are used as a baseline to show the improved predictive skill of ANN. The quantitative results would be more useful by including more information about that baseline.

p6,L11 (fig 4): How/where are the equal-area regions defined? Are those just pixel aggregated that meet the selection criteria for the sensitivity analysis?

p9, Fig 5. How is the CF and LWP sensitivity to AOD compatible? Is it a fair statement to say that we get more clouds with lower LWP for higher aerosol loading, while COD stays the same (perhaps because the "classical" indirect effect kicks in) - or can we not make such a blanket statement?

p10, L5: Would it make sense to plot co-sensitivity maps, considering that many predictands co-vary with predictors. In the inverse theory equivalent, one would consider

the off-diagonal elements of the covariance matrices. After all, one of the attractive features of this analysis is that it allows multi-variate analysis of ACI, fully considering the meteorologic conditions - but then the plots / analysis do not reap the full benefits of this approach. The authors do explain some of the co-variabilities/co-sensitivities, but then again it would be even better to have some graphical representation for some of these connections.

p10,L28: Does the CDR - AOD relationship for the SE Atlantic region make sense? For the outflow from the Arabian peninsula and the Sahara, it does, and the manuscript explains this with dust - but on the West coast of Namibia and Angola the dust is confined to the coast. It is possible that the identified relationships here points to limitations of the data set(s) that serve as the basis. Perhaps dust is overrepresented in the data? Overall, it would be good to see a discussion in which regions we would trust the correlations (given the uncertainties in the data).

p12, L15: So, cloud radiative effect sensitivities are actually not (yet) addressed in the manuscript. Instead, cloud properties are analyzed. Earlier in the manuscript (p4,L24), it is stated that cloud radiative effects are analyzed. This should be fixed (minor comment).

---

## Author Comment (AC3) · 12 Jun 2017

1 Understanding the drivers of marine liquid-water

2 cloud occurrence and properties with global

3 observations using neural networks

4 — RESPONSE TO REFEREE 3 —

5

6 contact: hendrik.andersen@kit.edu

We would like to thank referee 3 for her/his review of the manuscript and her/his constructive criticism. Comments by the referee are colored in blue, our replies are colored in black.

This paper pursues a promising approach to study the sensitivity of marine liquid-water cloud properties on a set of meteorological and aerosol predictors, using an artificial neural network approach. It steers clear of correlative approaches for studying aerosol-cloud interactions and instead considers the meteorological context, segregated by region / meteorological regime. In essence, this amounts to a multi-variate analysis based on an optimal combination of satellite and re-analysis data. The paper is very well written, clearly represents new ideas, and has the potential to lead to major improvements in our assessment of ACI, regionally and globally. It is rare to see such a high-quality paper. I only have minor comments, which don't necessarily have to be addressed in this manuscript, but could be considered in future work. The most important ones are probably #1 regarding scale, and regarding the quality (reliability) of the data. Also, follow-up papers might consider using the co-sensitivity of some predictors (details below).

In a separate comment to the editor, I recommended that the paper be highlighted because it seems highly innovative in its approach and deviates from the traditional correlative aerosol-cloud interaction studies. I believe that it has potential to change the direction of this field of research.

Thank you very much for this kind assessment. We respond to each point individually below.

**General comments:**

p5,L18: In the spirit of the McComiskey and Feingold ACI papers, it would have been interesting to also consider the impact of scale on ACI relationships. Here, one specific scale has been used (dictated by the analysis grid) - but it may not be straightforward to generalize these relationships.

This is a good point and we agree that the scale of the data sets used to study aerosol-cloud interactions influences the derived sensitivities (McComiskey et al., 2009; McComiskey and Feingold, 2012). Here, we use temporally and spatially highly aggregated data sets (monthly means in the defined equal-area regions), as with this study, we are specifically interested in the very large scale mechanisms and patterns of the aerosol-cloud-climate system. This is certainly not the scale at which the processes occur, so that our derived sensitivities may not match the magnitude of the sensitivities at the process scale. An analysis of the impact of the extent of spatial aggregation of the $1°x1°$ data on the derived sensitivities would be interesting; however, the spatial aggregation we chose was needed for sampling reasons (sufficient number of observations for the statistical model). In the revised version of the manuscript, we discuss this on P6L1–3. (*"As the temporal and spatial scales considered in this study are not on the same scale as the actual processes, so that the calculated sensitivities may not match the magnitude of the sensitivities at the process scale (McComiskey et al., 2009; McComiskey and Feingold, 2012)."*)

p6,L4: "skill of simple correlation between AOD & cloud properties": It is a bit unclear, which "simple correlations" specifically have been used for this study. This statement calls for elaboration. The statement on p6,L6/7 shows the intent - the "simple correlations" are used as a baseline to show the improved predictive skill of ANN. The quantitative results would be more

useful by including more information about that baseline.

Here, with "simple correlation" we referred to a "simple" Pearson correlation between AOD and either CLF/CDR/LWP/COT in each equal area region. In the revised version of the manuscript, we describe this at P6L8, however, in the current version of the manuscript, the results of Pearson correlations between log(AI) and the respective cloud properties is illustrated in figure 2.

p6,L11 (fig 4): How/where are the equal-area regions defined? Are those just pixel aggregated that meet the selection criteria for the sensitivity analysis? This is explained in the manuscript on P4L33-P5L3. The equal-area regions are defined by dividing the space between 60°N and 60°S (and all longitudes) into 20x40 equally sized areas. The original 1°x1° data is aggregated in these regions at their original spatial resolution. The selection criteria for the sensitivity analysis is checked for each equal-area region (but only for the sensitivity analysis - in figure 4, all equal-area regions are shown). In the revised version of the manuscript, we added some information to the caption of figure 4 for clarity.

p9, Fig 5. How is the CF and LWP sensitivity to AOD compatible? Is it a fair statement to say that we get more clouds with lower LWP for higher aerosol loading, while COD stays the same (perhaps because the "classical" indirect effect kicks in) - or can we not make such a blanket statement? The CLF sensitivity to AOD/AI is probably the sensitivity that is the most uncertain, due to cloud contamination of the satellite aerosol retrievals and the influence of confounding variables on both CLF and the satellite retrieved aerosol quantity. While we weaken the influence of confounding variables by including them in the ANN, we are not able to reduce effects related to data quality (this is discussed on P13L4–6 in the revised version of the manuscript:

[Figure]

Figure 1: Global map of LWP sensitivity to AI: The globally averaged sensitivities are based on the regions marked with a '+'.

"*While the influence of confounding factors is limited by the multivariate approach, effects concerning data quality (e.g. cloud contamination) are not accounted for and need to be considered when interpreting the CLF sensitivity to AI.*"). One should also note that the averaged LWP sensitivities rely on very few regions (due to the selection criteria) and should thus not be considered global. In most regions, the sensitivity of LWP to AI is relatively low.

While it makes sense to combine the sensitivities as proposed by you, one needs to remember that these are derived from separate ANNs. While LWP and CLF in the respective ANNs respond to AI/AOD in the way that you point out, changes in LWP might also affect CLF and vice versa, which would not be accounted for. Therefore, we are somewhat cautious in the interpretation of combined sensitivities.

p10, L5: Would it make sense to plot co-sensitivity maps, considering that many predictands co-vary with predictors. In the inverse theory equivalent, one would consider the off-diagonal elements of the covariance matrices. After all, one of the attractive features of this analysis is that it allows multi-variate

analysis of ACI, fully considering the meteorologic conditions - but then the plots / analysis do not reap the full benefits of this approach. The authors do explain some of the co-variabilities/co-sensitivities, but then again it would be even better to have some graphical representation for some of these connections. Yes, this is a good idea - and an idea which we discussed internally, as well. Ultimately, this level of detail exceeds the scope of this study, as one would have to create co-sensitivity plots for each grid-cell-specific ANN individually and would thus not be able to produce summarized global co-sensitivities easily. This is an idea we are currently pursuing in a more detailed regional study.

p10,L28: Does the CDR - AOD relationship for the SE Atlantic region make sense? For the outflow from the Arabian peninsula and the Sahara, it does, and the manuscript explains this with dust - but on the West coast of Namibia and Angola the dust is confined to the coast. It is possible that the identified relationships here points to limitations of the data set(s) that serve as the basis. Perhaps dust is overrepresented in the data? Overall, it would be good to see a discussion in which regions we would trust the correlations (given the uncertainties in the data).

This is a good question - in a regional study some years ago, we found that in certain conditions (stable/humid), AI and CDR are positively related in the Southeast Atlantic (Andersen and Cermak, 2015). However, in most cases, the AI-CDR relationship was found to be negative as in (e.g. Costantino and Bréon, 2013). This specific regional sensitivity may be affected by retrieval or sampling issues, as now discussed in the revised version of the manuscript on P10L5–8 (*"Issues of sampling (few aerosol retrievals in high CLF-regions) or scale (highly aggregated data) or their combination might affect the observed CDR sensitivity to AI in this region."*).

p12, L15: So, cloud radiative effect sensitivities are actually not (yet) ad-
dressed in the manuscript. Instead, cloud properties are analyzed. Earlier in
the manuscript (p4,L24), it is stated that cloud radiative effects are analyzed.
This should be fixed (minor comment).

Yes, you are correct. We have deleted the mentioned text passage in the revised
manuscript.

**References**

Andersen, H. and Cermak, J. (2015). How thermodynamic environments control
stratocumulus microphysics and interactions with aerosols. *Environmental
Research Letters*, 10(2):24004.

Costantino, L. and Bréon, F.-M. (2013). Aerosol indirect effect on warm clouds
over South-East Atlantic, from co-located MODIS and CALIPSO observa-
tions. *Atmospheric Chemistry and Physics*, 13(1):69–88.

McComiskey, A. and Feingold, G. (2012). The scale problem in quantifying
aerosol indirect effects. *Atmospheric Chemistry and Physics*, 12(2):1031–1049.

McComiskey, A., Feingold, G., Frisch, a. S., Turner, D. D., Miller, M. a., Chiu,
J. C., Min, Q., and Ogren, J. a. (2009). An assessment of aerosol-cloud
interactions in marine stratus clouds based on surface remote sensing. *Journal
of Geophysical Research: Atmospheres*, 114(D9):D09203.

---

## Editor Decision (ED1)

Referee #1:

The authors have done a lot of work going through my comments, which I appreciate.

One remaining issue is reconciling point (3) on page 5 of the author response document and (4) on page 6. I agree that it is interesting to look at these relationships with an NNA, but I don't understand how the complexity of the NNA is justified in this case. It seems that the system is relatively linear based on the new version of Figure 2. As the authors point out in (4) they are clarifying previous work, which I also think is a good thing to do, but if the NNA doesn't greatly alter the model skill beyond a multiple linear regression then it is more difficult to say that this is an independent test of the original results. To be publishable the paper needs to clarify what the NNA can do that the multiple linear regression cannot. I will admit that I am not very familiar with the literature on NNA, so I appreciate the authors taking time to explain what they are doing. Is there some criterion from the literature for when an NNA is an appropriate choice? Eg. something like Bayesian information criterion that would allow an objective statement of 'NNA is better than multiple linear regression'. If some criterion for NNA use from previous investigations can be met it seems like a reasonable analysis.

In regards to point 2 in the summary on page 12. Do the regional dependencies decrease if SST is added as a predictor? Between LTS, w, RH, BLH and SST that should encompass most of the factors that constitute a regime. However, I acknowledge that different factors should drive different regimes so the point that the authors are making seems reasonable.

Referee #2

I thank the authors for their work in responding to the comments. The majority of my comments have been addressed, only two things remain. I would suggest the authors include these at their own discretion. However, some mention of them might improve the paper still further.

The authors have been clear that they would prefer not to include results from a single global ANN. While creating a single relationship for predicting oceanic CLF globally may be difficult, this is not as far-fetched as they make it sound, as this is the central aim of a cloud parametrisation. By making their ANN regionally dependent, this is similar to including the latitude and longitude in a parametrisation. I understand that it is difficult to include all of the necessary parameters in a single ANN to predict the CLF (or other parameters), but as with a cloud parametrisation, the regions or conditions where the global ANN is deficient would indicate locations for future research.

Secondly, I think that my previous point about overlying ice cloud was misunderstood. The CLF is the fraction of retrievals where clouds with liquid tops are detected, so any situation where there is overlying ice cloud has the potential to reduce the CLF without changing the properties of the underlying liquid cloud. If there are not suitable variables in the ANN to predict the ice

cloud fraction, it essentially becomes random noise and would thus limit the predictive ability of the ANN. Using only single-layer cloud retrievals does not address this issue, as they provide only the fraction of retrievals where a single layer liquid water cloud is detected, a subset of CLF. The only way I can see to easily address this issue is to restrict the study to gridboxes where only liquid cloud is detected, ensuring that there is no overlying ice cloud to artificially reduce the CLF. While excluding pixels with ice clouds is not essential (as the ANN clearly already has significant predictive ability), I would recommend that the authors consider it as a method of improving the statistics generated by the ANN.

---

## Author Response (AR2)

**Understanding the drivers of marine liquid-water cloud occurrence and properties with global observations using neural networks**

**— Editor review —**

**— RESPONSE TO REFEREES 1 & 2 —**

contact: hendrik.andersen@kit.edu

We would like to thank referees 1 and 2 for their review of the revised manuscript and their constructive criticism. In the following, comments by the referees are colored blue, our replies are given in black.

**Referee 1**

The authors have done a lot of work going through my comments, which I appreciate.

One remaining issue is reconciling point (3) on page 5 of the author response document and (4) on page 6. I agree that it is interesting to look at these relationships with an NNA, but I don't understand how the complexity of the NNA is justified in this case. It seems that the system is relatively linear based on the new version of Figure 2. As the authors point out in (4) they are clarifying previous work, which I also think is a good thing to do, but if the NNA doesn't greatly alter the model skill beyond a multiple linear regression then it is more difficult to say that this is an independent test of the original results. To be publishable the paper needs to clarify what the NNA can do that the multiple linear regression cannot. I will admit that I am not very familiar with the literature on NNA, so I appreciate the authors taking time to explain what they are doing. Is there some criterion from the literature for when an NNA is an appropriate choice? Eg. something like Bayesian information criterion that would allow an objective statement of 'NNA is better than multiple linear regression'. If some criterion for NNA use from previous investigations can be met it seems like a reasonable analysis.

The novelty of our study is that we explicitly consider meteorological conditions together with aerosol to explain cloud properties in a multi-variate statistical approach. To do this, we chose an ANN for reasons explained below.

However, the choice of the ANN vs. other multi-variate methods is not the main innovation or benefit of this research.

We chose the ANN instead of the multiple linear regressions to analyze aerosol-cloud-climate interactions because:

1. "*Unlike other statistical techniques the multilayer perceptron makes no prior assumptions concerning the data distribution*" (Gardner and Dorling, 1998, p.2627).

2. "*It can model highly non-linear functions and can be trained to accurately generalise when presented with new, unseen data*" (Gardner and Dorling, 1998, p.2627).

Due to these properties, ANNs have generally been shown to better approximate relationships than multiple linear regressions (e.g. Sousa et al., 2007; Badr et al., 2014; Hartmann et al., 2016), especially when modeling nonlinear relationships (Badr et al., 2014), as is the case here. In Hartmann et al. (2016), the ANN is used, as its predictive capability is slightly higher than the multiple linear regression (as in our study). While there does not seem to be an objective criterion to decide when to use an ANN, typically the model with the highest predictive capability is chosen, which is the ANN in our case. More generally, Gardner and Dorling (1998, p.2627) state: "*These features [1 and 2 above] of the multilayer perceptron make it an attractive alternative to developing numerical models, and also when choosing between statistical approaches. As will be seen the multilayer perceptron has many applications in the atmospheric sciences*". Olden and Jackson (2002) state in their introduction that "*[t]he utility of ANNs for solving complex pattern recognition problems has been demonstrated in many terrestrial [sources] and aquatic studies [sources], and has led many researchers to advocate ANNs as an attractive, non-linear alternative to traditional statistical methods.*"

While we feel that the use of the ANNs in the context of aerosol-cloud-climate interactions is well justified, especially in light of the results, other multivariate statistical methods could have been applied as well, as shown in figure 1. We try to clarify these aspects in two sections of the revised version of the manuscript

P.2, L.16–17:"*The ANN is chosen as it is capable of modeling highly nonlinear functions and does not need any assumptions concerning the data distribution (Gardner and Dorling, 1998).* ".

P.6, L.12–13:"*While the ANN is chosen in this study due to its slightly superior predictive capabilities, figure 2 suggests that other multi-variate methods would have been appropriate as well.* ".

In regards to point 2 in the summary on page 12. Do the regional dependencies decrease if SST is added as a predictor? Between LTS, w, RH, BLH and SST that should encompass most of the factors that constitute a regime. However, I acknowledge that different factors should drive different regimes so the point that the authors are making seems reasonable.

To answer this question, we computed the ANNs including the SST as an additional predictor. The addition of SST to the ANNs did not meaningfully improve the predictive capability of the ANNs (c.f. fig. 1).

Also, regional dependencies are largely unchanged when SST is included. This is illustrated in fig. 2, where the panels show the global distributions of selected input-output sensitivities that feature strong (nearly identical) spatial patterns both without (left-hand panels) and with SST (right-hand panels) included in the model.

In our study design, we excluded SST from the model to prevent colinearity among the predictors (similar to using LTS and RH at various levels instead of

[Figure]

Figure 1: Model skills of ANNs, multiple linear regressions and bivariate (log(AI)- specific cloud property). For ANNs and multiple linear regressions, we included SST (right-hand panel), the version of the manuscript without SST is shown in the left-hand panel.

EIS), as a lot of the information content provided by SST is already indirectly included in the model via the surface temperatures used for the computation of LTS.

We have added SST as a potential predictor that could have additionally been included in the model on P.3, L.31.

**Referee 2**

I thank the authors for their work in responding to the comments. The majority of my comments have been addressed, only two things remain. I would suggest the authors include these at their own discretion. However, some mention of them might improve the paper still further.

The authors have been clear that they would prefer not to include results from a single global ANN. While creating a single relationship for predicting oceanic CLF globally may be difficult, this is not as far-fetched as they make it sound, as this is the central aim of a cloud parametrisation. By making their ANN regionally dependent, this is similar to including the latitude and longitude in a parametrisation. I understand that it is difficult to include all of the necessary

[Figure]

Figure 2: Patterns of the sensitivity of CDR to RH 850 (top) and LTS (bottom). Left-hand panels are computed without SST as in the manuscript, right-hand panels include SST as predictor in the ANN.

parameters in a single ANN to predict the CLF (or other parameters), but as with a cloud parametrisation, the regions or conditions where the global ANN is deficient would indicate locations for future research.

The cloud parameterization is an interesting analogy for using a single global ANN. When building such a single global ANN, our attempt would be to only use inputs that are physically related to the output. As such, incoming solar radiation at the surface could be a good approximation of the latitude. In our opinion, this is an interesting aspect for future research.

Yes, we agree that regions where our current ANN setup does not work quite as well can be viewed as locations where future research might help to find out which aspects of the system we are missing. This is an aspect that we are currently analyzing in a follow-up study. We mention this aspect now in the revised version of the manuscript (P.6, L.18–P.7, L.3):" *These regions with comparatively low ANN skill may point to regions where the aerosol-cloud-climate system cannot be sufficiently explained with the choice of predictors used in this study and may thus represent regions of interest for future studies.*".

Secondly, I think that my previous point about overlying ice cloud was misunderstood. The CLF is the fraction of retrievals where clouds with liquid tops are detected, so any situation where there is overlying ice cloud has the potential to reduce the CLF without changing the properties of the underlying liquid cloud. If there are not suitable variables in the ANN to predict the ice cloud fraction, it essentially becomes random noise and would thus limit the predictive ability of the ANN. Using only single-layer cloud retrievals does not address this issue, as they provide only the fraction of retrievals where a single layer liquid water cloud is detected, a subset of CLF. The only way I can see to easily address this issue is to restrict the study to gridboxes where only liquid cloud is detected, ensuring that there is no overlying ice cloud to artificially reduce the CLF. While excluding pixels with ice clouds is not essential (as the ANN clearly already has significant predictive ability), I would recommend that the authors consider it as a method of improving the statistics generated by the ANN.

You are right, we misunderstood this point. Yes, the liquid, single layer cloud fraction is the subset of the overall CLF where both liquid cloud tops and single layer situations are detected. As we are using monthly products, it is unclear how we would be able to exclude specific situations as suggested. While we agree that this aspect might weaken the predictive capability of the model, a large effect on the results of this study seems unlikely for two reasons:

1. Additionally filtering the data for single layer cloud situations (compared to the original manuscript that only filtered for liquid water clouds) did not alter the results meaningfully.

2. The situations described may average out to some extent in the long-term large-scale data sets used in this study.

Also, we are not quantitatively interpreting aerosol effects (e.g. in terms of radiative forcing computations), so that small effects do not meaningfully impact our results. We acknowledge and discuss your valid argument in the revised version of the manuscript (P.3, L.7–11):"*One should note that overlying ice-water clouds reduce the single-layer liquid-water cloud fraction, without actually changing the liquid-water cloud fraction below. This scenario would translate to random noise and potentially blur statistical relationships. However, these effects are thought to be minor, as these situations are likely to average out to some extent in the long-term large-scale data sets used in this study. Also, different cloud products where tested for this study that all yielded similar results.*".

**List of important changes**

P2 Included sentence for clarity on method selection (Referee 1)

P3 Additional discussion of the properties of the data sets (Referee 2)

P3 Mention of SST as a potential predictor (Referee 1)

P6 Included sentence for clarity on method selection (Referee 1)

P7 Discussion of regions with low ANN skill (Referee 2)

[revised manuscript text omitted]